# Fast Graph Generation via Autoregressive Noisy Filtration Modeling

**Markus Krimmel**                                                         *krimmel@biochem.mpg.de*
*Max Planck Institute of Biochemistry*

**Jenna Wiens**                                                              *wiensj@umich.edu*
*University of Michigan*

**Karsten Borgwardt**                                                     *borgwardt@biochem.mpg.de*
*Max Planck Institute of Biochemistry*

**Dexiong Chen**                                                           *dchen@biochem.mpg.de*
*Max Planck Institute of Biochemistry*

*Reviewed on OpenReview:* *https://openreview.net/forum?id=3Up8lZq728*

## Abstract

Existing graph generative models often face a critical trade-off between sample quality and generation speed. We introduce Autoregressive Noisy Filtration Modeling (ANFM), a flexible autoregressive framework that addresses both challenges. ANFM leverages filtration, a concept from topological data analysis, to transform graphs into short sequences of subgraphs. We identify exposure bias as a potential hurdle in autoregressive graph generation and propose noise augmentation and reinforcement learning as effective mitigation strategies, which allow ANFM to learn both edge addition and deletion operations. This unique capability enables ANFM to correct errors during generation by modeling non-monotonic graph sequences. Our results show that ANFM matches state-of-the-art diffusion models in quality while offering over 100 times faster inference, making it a promising approach for high-throughput graph generation. The source code is publicly available at https://github.com/BorgwardtLab/anfm.

## 1 Introduction

Graphs are fundamental structures that model relational data in various domains, from social networks and molecular structures to transportation systems and neural architectures. The ability to generate realistic and diverse graphs therefore holds great promise in many applications, such as drug discovery (Liu et al., 2018; Vignac et al., 2023), network simulation (Yu & Gu, 2019), and protein design (Ingraham et al., 2019). The space of drug-like molecules and protein conformations is, for practical purposes, infinite, limiting the effectiveness of in-silico screening of existing libraries (Polishchuk et al., 2013; Levinthal, 1969). Consequently, high-throughput graph generation—the task of efficiently creating new graphs that faithfully emulate properties similar to those observed in a given domain—is thus emerging as a critical challenge in machine learning and generative artificial intelligence.

Recent deep learning-based approaches, particularly autoregressive (You et al., 2018b; Liao et al., 2019; Kong et al., 2023) and diffusion models (Vignac et al., 2023; Bergmeister et al., 2024), have shown promise in generating increasingly realistic graphs. However, many current diffusion-based approaches rely on iterative refinement processes involving a large number of steps. This computational burden may hinder their potential for high-throughput applications (Gentile et al., 2022; Gómez-Bombarelli et al., 2016). While autoregressive models are more efficient during inference, they have underperformed in terms of generation

quality. Moreover, they might be susceptible to exposure bias (Ranzato et al., 2016), where performance deteriorates as errors accumulate during sampling and a train-test discrepancy consequently arises.

Recent work has explored the use of topological data analysis, particularly persistent homology and filtration (Edelsbrunner et al., 2002; Zomorodian & Carlsson, 2005), for graph representation. A filtration provides a multi-scale view of a graph structure by constructing a nested sequence of subgraphs. This approach has shown potential in various graph analysis tasks, including classification and similarity measurement (O'Bray et al., 2021; Schulz et al., 2022). In the context of generative modeling, filtration-based representations have been used to develop more expressive tools for generative model evaluation (Southern et al., 2023). However, the application of filtration-based methods for graph generation remains unexplored. In this work, we propose filtrations as a generalization of graph sequence families used in prior autoregressive models (You et al., 2018b; Liao et al., 2019), offering a flexible framework to construct sequences for generation. Nonetheless, modeling filtration sequences in a naive manner remains prone to exposure bias.

To address this, we introduce Autoregressive Noisy Filtration Modeling (ANFM), a novel approach to fast graph generation that models noise-augmented filtration sequences autoregressively. To generate a target graph, our method produces a short sequence of increasingly dense and detailed intermediate graphs, which interpolate the target graph and the fully disconnected graph. Compared to diffusion models (Vignac et al., 2023; Bergmeister et al., 2024), ANFM requires fewer iterations during sampling, resulting in significantly faster inference speed. By adding noise to the filtration sequences, ANFM learns to simultaneously remove or add edges. As a result, the model can recover from errors during sampling. Additionally, we further mitigate exposure bias with adversarial fine-tuning using reinforcement learning (RL). Our method offers a promising balance between efficiency and accuracy in graph generation, providing a 100-fold speedup over diffusion-based approaches, while substantially outperforming existing autoregressive models in terms of generation quality.

In summary, our contributions are as follows:

- We propose a novel autoregressive graph generation framework that leverages graph filtration. Our formulation generalizes the graph sequences used by previous autoregressive models that operate via node addition.
- We introduce a specialized autoregressive model architecture designed to operate on these graph sequences.
- We identify exposure bias as a potential challenge in autoregressive graph generation and propose noise augmentation and adversarial fine-tuning as effective strategies to mitigate this issue.
- We conduct ablation studies to evaluate the impact of different components within our framework, demonstrating that noise augmentation and adversarial fine-tuning substantially improve performance.
- Our empirical results highlight the strong performance and efficiency of our model compared to recent baselines. Notably, our model achieves inference speed 100 times faster than existing diffusion-based models.

## 2 Related Work

ANFM builds on the concept of graph filtration and incorporates noise augmentation. It is fine-tuned via reinforcement learning to mitigate exposure bias. In the following, we provide a brief overview of related graph generative models (GGMs), approaches to address exposure bias, and applications of graph filtration.

**Autoregressive GGMs.** GraphRNN (You et al., 2018b) made the first advances towards deep generative graph models by autoregressively generating nodes and their incident edges to build up an adjacency matrix row-by-row. In a similar fashion, DeepGMG (Li et al., 2018) iteratively builds a graph in a node-by-node fashion. Liao et al. (2019) proposed a more efficient autoregressive model, GRAN, by generating multiple nodes at a time in a block-wise fashion, leveraging mixtures of multivariate Bernoulli distributions. GraphArm (Kong et al., 2023) introduced an autoregressive model that reverses a diffusion process in which nodes and their incident edges decay to an absorbing state. These models share the property that they build graphs by node addition. Hence, they autoregressively generate an increasing sequence of *induced subgraphs*

(i.e., maximal subgraphs on a subset of the nodes). In comparison, the subgraphs we consider in our work do not necessarily need to be induced. Moreover, ANFM may generate sequences that are not monotonic. That is, ANFM may simultaneously add and *delete* edges. In contrast to autoregressive node-addition methods, approaches by Goyal et al. (2020) and Bacciu et al. (2020) generate graphs through edge-addition following a pre-defined edge ordering.

**Diffusion GGMs.** Diffusion models for graphs such as EDP-GNN (Niu et al., 2020) and GDSS (Jo et al., 2022), based on score matching, or DiGress (Vignac et al., 2023), based on discrete denoising diffusion (Austin et al., 2021), have emerged as powerful generators. However, they require many iterative denoising steps, making them slow during sampling. Hierarchical approaches (Bergmeister et al., 2024) and absorbing state processes (Chen et al., 2023) have subsequently been proposed to allow diffusion models to be scaled to large graphs. In contrast to the noise processes in denoising diffusion models, the filtration processes we consider are in general *non-Markovian*, necessitating a full autoregressive modeling.

In concurrent work, Boget (2025) proposed SID, a modification of discrete denoising graph diffusion in which intermediate states are conditionally independent. This work was motivated by a phenomenon similar to exposure bias, termed *compounding denoising errors*. Similar to ANFM, the sequences modeled by an absorbing state variant of SID are non-montonic and non-Markovian.

**Single-Step GGMs.** Unlike iterative approaches such as autoregression and diffusion, graph variational autoencoders (VAEs) (Kipf & Welling, 2016; Simonovsky & Komodakis, 2018) generate all edges in a single step, reducing computational costs during inference. However, VAEs struggle to model complicated distributions and require graph matching during training, restricting their application to small graphs. Generative adversarial networks (GANs) (Goodfellow et al., 2014) offer an alternative, operating in a likelihood-free fashion and avoiding graph matchings. Despite this advantage, GANs are notoriously difficult to train and suffer from issues such as mode collapse (Cao & Kipf, 2018). To address these instabilities, Martinkus et al. (2022) proposed SPECTRE, a GAN that first generates a Laplacian spectrum before producing the corresponding graph.

**RL in Graph Generation.** In the context of graph generation, reinforcement learning has been used mainly to generate molecular graphs. You et al. (2018a) trained a generative model for molecules via RL, combining adversarial and domain-specific rewards. In contrast to our work, they only considered molecular graphs and did not use teacher-forcing during training. Taiga (Mazuz et al., 2023) used reinforcement learning to optimize the chemical properties of molecules obtained from a language model pre-trained on SMILES strings. Diffusion models have also been shown to be amenable to RL finetuning, allowing extrinsic non-differentiable metrics to be optimized (Liu et al., 2024).

**Exposure Bias.** Exposure bias (Bengio et al., 2015; Ranzato et al., 2016) refers to the train-test discrepancies autoregressive models face when they are exposed to their own predictions during inference. Errors may accumulate during sampling, leading to a distribution shift and degrading performance. To mitigate this phenomenon in natural language generation, Bengio et al. (2015) proposed a data augmentation strategy to expose models to their predictions during training. In a similar effort, Ranzato et al. (2016) proposed training in free-running mode using reinforcement learning to optimize sequence-level performance metrics. SeqGAN (Yu et al., 2017), which is the most relevant to our work, also trains models in free-running mode using reinforcement learning. Instead of relying on extrinsic metrics like Ranzato et al. (2016), it adversarially trains a discriminator to provide feedback to the generative model. GCPN (You et al., 2018a) adopts a hybrid framework for generating small molecules, combining adversarial and domain-specific rewards.

**Graph Filtration.** Filtration is commonly used in the field of persistent homology (Edelsbrunner et al., 2002) to extract features of geometric data structures at different resolutions. Previously, graph filtration has mostly been used to construct graph kernels (Zhao & Wang, 2019; Schulz et al., 2022) or extract graph representations that can be leveraged in downstream tasks such as classification (O'Bray et al., 2021). While filtration has also been used for evaluating graph generative models (Southern et al., 2023), *to the best of our knowledge, our work presents the first model that directly leverages filtration for generation.*

## 3    Background

In the following, we consider unlabeled and undirected graphs, denoted by $G = (V, E)$, where $V$ is the set of vertices and $E \subseteq V \times V$ is the set of edges. Without loss of generality, we assume $V = \{1, 2, \ldots, n\}$ and denote by $e_{ij}$ the edge between nodes $i, j \in V$. We assume that only connected graphs are presented to our model during training and filter training sets if necessary. Our approach is based on the concept of graph filtration.

**Graph Filtration.**    A filtration of a graph $G$ is defined as a nested sequence of subgraphs:

$$G = G_T \supseteq G_{T-1} \supseteq \cdots \supseteq G_1 \supseteq G_0 = (V, \emptyset) \tag{1}$$

where each $G_t = (V, E_t)$ is a graph sharing the same node set as $G_T := G$. The filtration satisfies the following properties: (1) $E_t \subseteq E_{t'}$ for all $t < t'$ and (2) $G_0$ is the fully disconnected graph, *i.e.,* $E_0 = \emptyset$. The hyper-parameter $T$ controls the length of the sequences, which is selected to be typically small in our experiments ($T \leq 32$).

**Filtration Function and Schedule.**    A convenient method to define a filtration of $G$ involves specifying two key components (O'Bray et al., 2021): a *filtration function* defined on the edge set $f : E \to \mathbb{R}$ and a non-decreasing sequence of scalars $(a_0, a_1, \ldots, a_T)$ with $-\infty = a_0 \leq a_1 \leq \cdots \leq a_{T-1} \leq a_T = +\infty$. Given these components, we can define the edge sets $E_t$ as nested sub-levels of the function $f$ for $t = 1, \ldots, T-1$.:

$$E_t := f^{-1}((-\infty, a_t]) = \{e \in E \ : \ f(e) \leq a_t\} \tag{2}$$

The sequence $(a_t)_{t=0}^T$ is referred to as the *filtration schedule sequence*. The choice of the filtration function and the schedule sequence plays a crucial role in effective graph generation. We present a visual example of the filtration process in Figure 1b.

## 4    Autoregressive Noisy Filtration Modeling

In this section, we present the Autoregressive Noisy Filtration Modeling (ANFM) approach for graph generation. Given a node set $V$, our objective is to generate a sequence of increasingly dense graphs $\tilde{G}_0, \tilde{G}_1, \ldots, \tilde{G}_T$ on $V$. The final graph $\tilde{G}_T$ should plausibly represent a sample from the target data distribution. To achieve this goal, ANFM will be trained to reverse a noise augmented filtration process.

This section is organized as follows: We present two filtration strategies in Sec. 4.1. To mitigate exposure bias, we propose a noise-augmentation of the resulting graph sequences in Sec. 4.2. We then introduce in Sec. 4.3 our autoregressive model that reverses this noisy filtration process. In Sec. 4.4, we propose a two-staged training scheme for ANFM, introducing an adversarial fine-tuning stage to further address exposure bias. Finally, in Sec. 4.5 we discuss algorithmic differences of ANFM and existing graph generative models.

As our proposed method consists of several components, we study their individual contributions in extensive ablation experiments in Sec. 5.4 and Section H. Furthermore, implementation details and hyperparameter settings are provided in Sections A, D, E and J.

### 4.1    Filtration Strategies

In the following, we discuss two primary strategies for the filtration function and schedule. Alternative choices are investigated in Appendix H.

**Filtrations from Node Orderings.**    Many existing autoregressive models operate via node addition and thereby model sequences of nested induced subgraphs (You et al., 2018b; Li et al., 2018; Liao et al., 2019; Kong et al., 2023). We show that similar sequences may be obtained via filtration. In this sense, the filtration framework generalizes the sequences considered by these previous works. Given a graph $G$, let $g : V \to \{1, \ldots, n\}$ be a node ordering, *i.e.,* a bijection. Models that operate via node addition generate a graph sequence

$$(\emptyset, \emptyset) = G[V_0] \subseteq \cdots \subseteq G[V_T] = G, \tag{3}$$

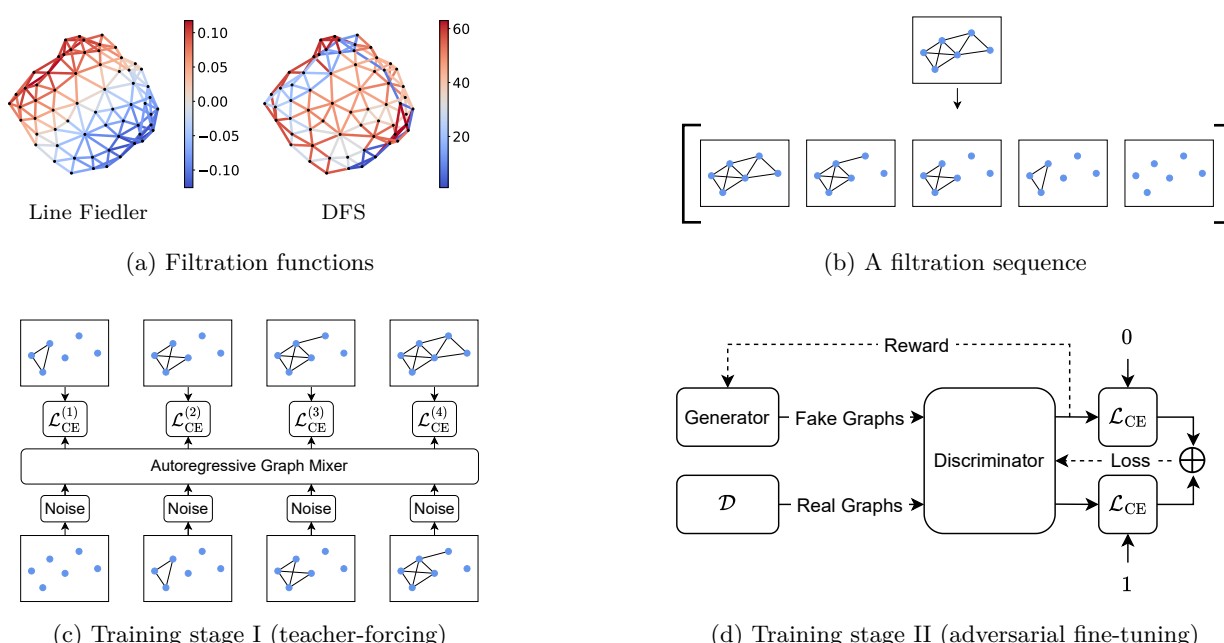

(a) Filtration functions

(b) A filtration sequence

(c) Training stage I (teacher-forcing)

(d) Training stage II (adversarial fine-tuning)

Figure 1: Top: A graph is transformed into a sequence of subgraphs (filtration) by taking sub-levels of a filtration function (i.e., performing edge-deletion according to the order prescribed by the filtration function). Bottom left: the generator is trained via teacher-forcing to reverse the filtration process. Bottom right: the generator is fine-tuned in free-running mode via reinforcement learning on a reward signal output by a discriminator in a SeqGAN-like framework (c.f. Appendix J).

where $V_0, \ldots, V_T$ are monotonically increasing sub-levels of $g$ with $V_t = g^{-1}([(-\infty, a_t])$ for some scalars $-\infty = a_0 \leq \cdots \leq a_T = +\infty$, and $G[V_t]$ denotes the induced subgraph whose node set is $V_t$. Now we consider the following filtration function $f : E \to \mathbb{R}$:

$$f(\{u, v\}) := \max\{g(u), g(v)\} \qquad \forall \{u, v\} \in E. \tag{4}$$

It is not hard to verify that this filtration function combined with the filtration schedule $(a_t)_{t=0}^T$, yields a filtration sequence $(G_t)_{t=0}^T$ in which the edge set of $G_t$ coincides with the edge set of $G[V_t]$ for any $t = 0, \ldots, T$. Hence, this filtration closely mirrors the sequence of induced subgraphs, differing only in that the node set does not change over time. We say that graphs in this sequence have *induced edge sets*. While a filtration function $f$ may be derived from any node ordering, we focus on depth-first search (DFS) orderings in this work, as shown to be optimal among several orderings for autoregressive graph generation in GRAN (Liao et al., 2019). We visualize resulting edge weights on a planar graph in Figure 1a. For filtrations derived from DFS orderings, we choose the filtration schedule $a_t$ to linearly increase from a minimum edge weight at $t = 1$ ($a_1 = 2$) to a maximum edge weight at $t = T$ ($a_T = |V|$). Unlike in previous autoregressive models (You et al., 2018b; Liao et al., 2019; Kong et al., 2023), $T$ is fixed across all graphs. We argue that a fixed $T$ simplifies both training and inference: since sequence lengths are uniform across all graphs, no padding must be applied during training. Additionally, no end-of-sequence signal must be emitted during inference.

**Line Fiedler Filtrations.** In contrast to the family of graph sequences leveraged by node-addition approaches, the filtration framework is not limited to sequences of induced subgraphs. We present the *line Fiedler* filtration function, which is one particular choice that, in general, yields a sequence of subgraphs with non-induced edge sets. This filtration function is intended to decompose graphs in a local fashion, assigning similar edge weights to neighboring edges. We are motivated by the intuition that it may potentially be beneficial to construct graphs by adding edges that are incident to the same vertex at similar timesteps in the generation process. To achieve this goal, we leverage the first non-trivial eigenvectors of graph Laplacians, called *Fiedler vectors*. It is known that these eigenvectors represent low-frequency signals on vertices suitable

for clustering (Shi & Malik, 1997) and often coloring them in a "continuous gradient". Since we are interested in coloring the *edges* of a given graph $G$, we transition to its line graph (Harary & Norman, 1960) $L(G)$. Edges in $G$ form vertices in $L(G)$, and they are adjacent in $L(G)$ if they are incident to a shared vertex in $G$. The Fiedler vector of $L(G)$ now tends to color its vertices in a continuous gradient, which transfers to a continuous coloring of the edges of $G$. We choose the line Fiedler filtration function as such a Laplacian eigenvector of $L(G)$ and refer to Appendix M for a more rigorous definition. We visualize the resulting edge weights on a planar graph in Figure 1a, demonstrating how it varies across edges in a "continuous gradient". For $0 \leq t \leq T$, we propose to define the filtration schedule $a_t$ as:

$$a_t := \inf \left\{ a \in \mathbb{R} \ : \ \frac{|f^{-1}((-\infty, a_t])|}{|E|} \geq \gamma(t/T) \right\}, \tag{5}$$

where $f$ is the line Fiedler filtration function, and $\gamma : [0, 1] \to [0, 1]$ is a monotonic function governing the rate at which edges are added in the filtration sequence. We choose $\gamma(t) := t$, leading to an approximately linear increase in the number of edges throughout the graph sequence. We investigate other choices of $\gamma$ in Appendix H.

## 4.2   Noise Augmentation of Filtrations

Our goal is to autoregressively generate sequences of graphs that approximately reverse the filtration processes above. To mitigate exposure bias in autoregressive modeling, previous works have proposed data augmentation schemes to make models more robust to the distribution shift occurring during inference (Bengio et al., 2015). We propose a simpler yet effective strategy: namely, randomly perturbing intermediate graphs in a filtration sequence $G_0, \ldots, G_T$ during the first training stage to expose the model to erroneous transitions. For each intermediate graph $G_t$ with $0 < t < T$, we generate a perturbed graph $\tilde{G}_t$ with edge set $\tilde{E}_t$ by including each possible edge $e$ independently with probability

$$\mathbb{P}[e \in \tilde{E}_t] := \begin{cases} (1 - \lambda_t) + \lambda_t \rho_t & \text{if} \quad e \in E_t \\ \lambda_t \rho_t & \text{else} \end{cases}, \tag{6}$$

where $\lambda_t \in [0, 1]$ controls stochasticity and $\rho_t := |E_t| / \binom{|V|}{2}$ is the density of $G_t$. In practice, we decrease $\lambda_t$ linearly as $t$ increases and include multiple perturbations of each filtration sequence in the training set. The choice of hyper-parameters, such as $\lambda_t$ and the number of included perturbations, is detailed in Appendix D.2. We note that this augmentation yields non-monotonic noisy filtration sequences $(\tilde{G}_t)_{t=0}^{T}$ during training. Hence, our autoregressive model is trained to allow for edge deletions.

## 4.3   Autoregressive Modeling of Graph Sequences

The generative model will be trained to reverse the noise-augmented filtration process detailed above. We formulate the generative process using an autoregressive model, expressing the joint likelihood as follows:

$$p_\theta(\tilde{G}_T, \ldots, \tilde{G}_0) = p(\tilde{G}_0) \prod_{t=1}^{T} p_\theta(\tilde{G}_t | \tilde{G}_{t-1}, \ldots, \tilde{G}_0), \tag{7}$$

where $p(\tilde{G}_0)$ represents the distribution over initial graphs, defined as a point mass on the fully disconnected graph $(V, \emptyset)$. In the following, we will detail our implementation of the autoregressive model $p_\theta$, including the architecture and training procedure. While existing autoregressive models typically utilize RNNs (You et al., 2018b; Liao et al., 2019; Goyal et al., 2020; Bacciu et al., 2020) or a first-order autoregressive structure (Kong et al., 2023), our model architecture for implementing $p_\theta$ is a novel and efficient design inspired by MLP-Mixers (Tolstikhin et al., 2021).

**Backbone Architecture.**   The graph sequences we consider can be viewed as dynamic graphs with constant node sets but evolving edge sets. Our backbone architecture operates on this structure by alternating between two types of information processing layers. The first type, called structural mixing, consists of a GNN that processes graph structures $\tilde{G}_0, ..., \tilde{G}_{T-1}$ independently, with weights shared across time steps.

The second type, called temporal mixing, consists of a transformer decoder (TRDecoder) that processes node representations along the temporal axis, with weights shared across nodes. Our model inherits the causal structure of the transformer decoder, ensuring that node representations at timestep $t$ only depend on the graphs $\tilde{G}_0, ..., \tilde{G}_t$. Formally, given input node representations $v_i^{(t)} \in \mathbb{R}^D$ for nodes $i \in V$ and time steps $t \in [T-1]$, a single mixing operation in our backbone model produces new representations $u_i^{(t)}$ and is defined as:

$$
\begin{aligned}
\left(w_i^{(t)}\right)_{i=1}^{|V|} &:= \text{GNN}_\theta \left( \left(v_i^{(t)}\right)_{i=1}^{|V|}, \tilde{E}_t, t \right) & \forall\, t = 0, ..., T-1, \\
\left(u_i^{(t)}\right)_{t=0}^{T-1} &:= \text{TRDecoder}_\theta \left( \left(w_i^{(t)}\right)_{t=0}^{T-1} \right) & \forall\, i = 1, ..., |V|,
\end{aligned}
\tag{8}
$$

where the first equation defines a structural mixing operation and the second equation defines a temporal mixing operation. For the structural mixing, we use Structure-Aware-Transformer layers (Chen et al., 2022). Additionally, we incorporate both the timestep $t$ and cycle counts in $\tilde{G}_t$ using FiLM (Perez et al., 2018). These structural features were used previously in other graph generative models such as DiGress (Vignac et al., 2023). Multiple mixing operations are stacked to form the backbone model.

**Edge Decoder.** To model $p_\theta(\tilde{G}_t | \tilde{G}_{t-1}, \ldots, \tilde{G}_0)$, we produce a distribution over possible edge sets of $\tilde{G}_t$. We use a mixture of multivariate Bernoulli distributions to capture dependencies between edges, similar to previous works (Liao et al., 2019; Kong et al., 2023). Mixture distributions are more expressive than simple multivariate Bernoulli distributions, which model the existence of edges independently in each generation step. Given $K \geq 1$ mixture components, we infer $K$ Bernoulli parameters for each node pair $i, j \in V$ from the node representations $v_i$ produced by the backbone model at timestep $t-1$:

$$
p_k^{(i,j)} := D_{k,\theta}(v_i, v_j) \in [0, 1],
\tag{9}
$$

where $k = 1, ..., K$ and $D_{\cdot,\theta}$ is some neural network. We enforce that $p_k^{(i,j)}$ is symmetric and that the probability of self-loops is zero. In addition, we produce a mixture distribution $\pi \in \mathbb{R}^K$ in the $K-1$ dimensional probability simplex from pooled node representations. The architectural details of $D_{\cdot,\theta}$ are provided in Appendix E.3. The final likelihood is defined as:

$$
p_\theta(\tilde{E}_t | \tilde{G}_{t-1}, ..., \tilde{G}_0) := \sum_{k=1}^{K} \pi_k \prod_{i<j} \left\{ \begin{array}{l} p_k^{(i,j)} \text{ if } e_{ij} \in \tilde{E}_t \\ 1 - p_k^{(i,j)} \text{ else} \end{array} \right\}
\tag{10}
$$

In contrast to existing autoregressive graph generators (You et al., 2018b; Liao et al., 2019; Goyal et al., 2020; Bacciu et al., 2020; Kong et al., 2023), *our model introduces a key innovation: the ability to generate non-monotonic graph sequences.* This means it can both add and delete edges. We argue that this capability is crucial for mitigating error accumulation during sampling (i.e., exposure bias). Consider, for instance, the task of generating tree structures. If a cycle is inadvertently introduced into an intermediate graph $\tilde{G}_t$ (where $t < T$), traditional autoregressive approaches would be unable to rectify this error. Our model, however, can potentially delete the appropriate edges in subsequent timesteps, thus recovering from such mistakes. The noise augmentation approach from Sec. 4.2 exposes ANFM to such erroneous transitions during training. We show empirically in Sec. 5.4 that this augmentation substantially improves performance.

**Input Node Representations.** The initialization of node representations is a crucial step preceding the forward pass through the mixer architecture above. We compute initial node representations from positional and structural features in a similar fashion as Vignac et al. (2023). Moreover, we add learned positional embeddings based on a node ordering derived from the filtration function. We refer to Section E.1 for further implementational details and to Section H for ablations of the node ordering.

**Asymptotic Complexity.** We provide a detailed analysis of the asymptotic runtime complexity of our method in Appendix B.1. Asymptotically, ANFM's complexity of sampling a graph with $N$ nodes is $\mathcal{O}(T^2 N + T N^3)$, where we recall that $T$ denotes the number of filtration steps. Although cubic in the number of nodes, we found that the efficiency of ANFM is largely driven by our ability to use a small $T$ ($T \leq 32$), while diffusion-based models generally require a much larger number of iterations.

While ANFM is in practice faster during inference than competing methods, we find that training could be more expensive. In Appendix K, we compare the training costs of ANFM to those of DiGress.

### 4.4 Training Algorithm

**Teacher-Forcing.** We employ teacher-forcing (Williams & Zipser, 1989) to train our generative model $p_\theta$ in a first training stage. We illustrate this training scheme in Figure 1c. Teacher-forcing allows the model to learn from complete sequences of graph evolution, providing a good initialization for subsequent reinforcement learning-based fine-tuning. Given a dataset of graphs $\mathcal{D}$, we convert it into a dataset of noisy filtration sequences, denoted as $\tilde{\mathcal{D}}$. Our objective is to maximize the log-likelihood of these sequences under our model:

$$\mathcal{L}(\theta) := \mathbb{E}_{(\tilde{G}_0,\ldots,\tilde{G}_T)\sim\tilde{\mathcal{D}}} \left[ \log p_\theta(\tilde{G}_0,\ldots,\tilde{G}_T) \right]. \tag{11}$$

In practice, this objective is implemented as a cross-entropy loss. While the noise augmentation introduced in Sec. 4.2 improves the overall quality of generated graphs after teacher-forcing training, it still falls short in generating graphs with high structural validity. To further mitigate exposure bias, we propose an RL-based fine-tuning stage to refine the model trained with teacher-forcing.

**Adversarial Fine-tuning with RL.** Adapting the SeqGAN framework (Yu et al., 2017), we implement a generator-discriminator architecture where the generator (our mixer model) operates in inference mode as a stochastic policy and is thereby exposed to its own predictions during training. The discriminator is a graph transformer, namely GraphGPS (Rampásek et al., 2022). During training, the generator produces graph samples, which the discriminator evaluates for plausibility. The generator is updated using Proximal Policy Optimization (PPO) (Schulman et al., 2017) based on the discriminator's feedback, while the discriminator is trained adversarially to distinguish between generated and training set graphs. This training scheme is illustrated in Figure 1d. It is worth noting that only the final generated graph is presented to the discriminator. Therefore, the generator is trained to maximize a terminal reward without constraints on intermediate graphs. We provide pseudo-code in Appendix J. While we focus on adversarial fine-tuning of ANFM, similar techniques might prove useful for diffusion models or existing autoregressive graph generators. However, incorporating this training strategy into existing generative models would require substantial modifications, falling outside the scope of our work.

### 4.5 Comparison to Other Generation Paradigms

While we categorize ANFM as an autoregressive model, it also exhibits superficial similarities to diffusion-based approaches. Hence, we further clarify its conceptual similarities and differences to existing generation paradigms.

**Autoregressive Approaches.** Existing autoregressive graph generators build graphs iteratively by generating sequences of node-addition (You et al., 2018b; Liao et al., 2019; Li et al., 2018; Kong et al., 2023) or edge-addition (Goyal et al., 2020; Bacciu et al., 2020) operations, thus modeling *monotonic* sequences of subgraphs. The operations performed by these models are non-reversible, making them vulnerable to exposure bias. In contrast, ANFM explicitly models sequences of *entire graphs*, allowing for both edge addition and edge *deletion*. ANFM is encouraged to perform both operations by training with noise augmentations, thus allowing ANFM to generate *non-monotonic* graph sequences. We demonstrate in Sec. 5.4 that noise augmentation improves performance substantially, justifying the necessity of modeling non-monotonic sequences. Graph filtration is a natural and general framework for defining *monotonic* graph sequences to which we apply noise augmentations during the first training stage. This approach generalizes existing autoregressive methods by allowing sequences of subgraphs with non-induced edge sets to be constructed. While this enables the exploration of a wider variety of graph sequences for training, the non-induced sequences we investigate in our experiments (c.f., Sec. 5) are oftentimes outperformed by sequences of subgraphs with induced edge sets.

In autoregressive tasks, one is usually interested in the entire generated sequence. In contrast, we model the sequence $\tilde{G}_0,\ldots,\tilde{G}_T$ but are only interested in the marginal of $\tilde{G}_T$. This is akin to diffusion modeling,

where intermediate states serve as latent variables. Hence, we now clarify why we do not categorize ANFM as a diffusion model.

**Diffusion Models.** Similar to graph denoising diffusion models (Vignac et al., 2023; Chen et al., 2023; Kong et al., 2023), we reverse a corrupting process that transforms graph samples $G_T$ into graphs $\tilde{G}_0$ from a convergent distribution. Unlike standard denoising diffusion models, our filtration process is explicitly *non-Markovian*. Instead of simply accumulating noise over time steps, our sequence $\tilde{G}_T, \ldots, \tilde{G}_0$ is based on the topology of $G_T$. Hence, it may not be Markov. This is why we introduce a full autoregressive structure, which leads to improvements over a first-order structure (used in diffusion models), as we demonstrate in Appendix B.2. In practice, this filtration-based approach allows us to perform substantially fewer generation steps than diffusion models such as DiGress (Vignac et al., 2023), EDGE (Chen et al., 2023), or GraphARM (Kong et al., 2023).

## 5 Experiments

We empirically evaluate our method on synthetic and real-world datasets. We investigate the filtration strategy based on depth-first search node orderings (DFS) and the line Fiedler function (Fdl.). In Sec. 5.1, we first present results on the commonly used small benchmark datasets (Martinkus et al., 2022), comparing our method to a variety of baselines. We then demonstrate in Sec. 5.2 that we can improve upon these results by using a more realistic setting with more training examples. Additionally, we present results for inference efficiency. Finally, in Sec. 5.3, we demonstrate that our model is applicable to real-world data, namely larger protein graphs (Dobson & Doig, 2003) and drug-like molecules (Brown et al., 2019). In Sec. 5.4, we present ablation studies demonstrating the efficacy of noise augmentation and adversarial fine-tuning.

**Evaluation.** We follow established practices from previous works (You et al., 2018b; Martinkus et al., 2022; Vignac et al., 2023) in our evaluation. We compare a set of model-generated samples to a test set via maximum mean discrepancy (MMD) (Gretton et al., 2012), based on various graph descriptors. These descriptors include histograms of node degrees (Deg.), clustering coefficients (Clus.), orbit count statistics (Orbit), and eigenvalues (Spec.).

In previous works (Martinkus et al., 2022; Vignac et al., 2023), very few samples are generated for the evaluation of graph generative models. In Appendix I, we show theoretically and empirically that this leads to high bias and variance in the reported metrics. In Sec. 5.2 and 5.3, we generate 1024 samples for evaluation to mitigate this, while we generate 40 samples in Sec. 5.1 to fairly compare to previous methods. For synthetic datasets, we follow previous works by reporting the ratio of generated samples that are valid, unique, and novel (VUN). In Sec. 5.2 and 5.3, we report inference speed, measured as the time needed to generate 1024 graphs on an H100 GPU, normalized to a per-graph cost. Hence, the reported metrics have the unit second/graph. The measured inference times inherently depend on the implementation details of the methods. While we ensure that all models are evaluated on identical hardware under comparable conditions, this limitation cannot be fully eliminated. For each evaluation metric, we highlight the best and second-best performing method in bold and underlined, respectively.

**Baselines.** We aim to demonstrate that our method is competitive with state-of-the-art diffusion models in terms of sample quality while outperforming them in terms of inference speed. Hence, we compare our method to two recent diffusion models, namely DiGress (Vignac et al., 2023) and ESGG (Bergmeister et al., 2024). DiGress first introduced discrete diffusion to the area of graph generation and remains one of the most robust baselines. ESGG is acutely relevant to our work, as it aims to improve inference speed. In addition, we also present results on an autoregressive model, GRAN (Liao et al., 2019), which focuses on efficiency during inference. For details on model selection and hyper-parameters for these baselines, we refer to Sections G.1 to G.5. In Sec. 5.1, we report baseline results from the literature, also comparing to the hierarchical HiGen (Karami, 2024) approach, the scalable EDGE (Chen et al., 2023) diffusion model, the autoregressive GraphRNN model (You et al., 2018b), and the GAN-based SPECTRE model (Martinkus et al., 2022).

## 5.1 Experiments with Small Synthetic Datasets

Table 1: Performance of models on small synthetic SPECTRE datasets. Results on GraphRNN, GRAN and SPECTRE taken from Martinkus et al. (2022). Results on DiGress, ESGG and EDGE from Bergmeister et al. (2024).

| | Planar Graphs ($|V| = 64$, $N_{\text{train}} = 128$) | | | | | SBM Graphs ($|V| \sim 104$, $N_{\text{train}} = 128$) | | | | |
|---|---|---|---|---|---|---|---|---|---|---|
| | VUN ($\uparrow$) | Deg. ($\downarrow$) | Clus. ($\downarrow$) | Orbit ($\downarrow$) | Spec. ($\downarrow$) | VUN ($\uparrow$) | Deg. ($\downarrow$) | Clus. ($\downarrow$) | Orbit ($\downarrow$) | Spec. ($\downarrow$) |
| GraphRNN | 0.0 | 0.0049 | 0.2779 | 1.2543 | 0.0459 | 5.0 | 0.0055 | 0.0584 | 0.0785 | 0.0065 |
| GRAN | 0.0 | 0.0007 | 0.0426 | **0.0009** | 0.0075 | 25.0 | 0.0113 | 0.0553 | 0.0540 | 0.0054 |
| SPECTRE | 25.0 | 0.0005 | 0.0785 | 0.0012 | 0.0112 | 52.5 | 0.0015 | 0.0521 | 0.0412 | 0.0056 |
| DiGress | 77.5 | 0.0007 | 0.0780 | 0.0079 | 0.0098 | 60.0 | 0.0018 | **0.0485** | 0.0415 | **0.0045** |
| EDGE | 0.0 | 0.0761 | 0.3229 | 0.7737 | 0.0957 | 0.0 | 0.0279 | 0.1113 | 0.0854 | 0.0251 |
| HiGen | - | - | - | - | - | - | 0.0019 | 0.0498 | 0.0352 | **0.0046** |
| ESGG | **95.0** | 0.0005 | 0.0626 | 0.0017 | 0.0075 | 45.0 | 0.0119 | 0.0517 | 0.0669 | 0.0067 |
| ANFM (Fdl.) | 72.5 | 0.0037 | 0.1332 | 0.0047 | 0.0099 | 47.5 | 0.0014 | 0.0506 | 0.0551 | 0.0058 |
| ANFM (DFS) | 37.5 | **0.0004** | **0.0309** | 0.0012 | **0.0061** | **65.0** | **0.0007** | 0.0488 | **0.0335** | 0.0048 |

As a first demonstration of our method, we present results on the planar and SBM datasets by Martinkus et al. (2022). Since the training set consists of only 128 graphs, we find that ANFM models using the line Fiedler function tend to overfit during the teacher-forcing training stage. To mitigate this issue, we introduce some small stochastic perturbations to node orderings used for initializing node representations. We discuss this in more detail in Appendix E.1. Model selection is performed based on the minimal validation loss. Table 1 illustrates that, in terms of VUN on the planar graph dataset, our models outperform GraphRNN (You et al., 2018b), GRAN (Liao et al., 2019), EDGE Chen et al. (2023), and SPECTRE (Martinkus et al., 2022). While they perform worse than DiGress (Vignac et al., 2023) and ESGG (Bergmeister et al., 2024) on the planar graph dataset in terms of VUN, the DFS variant performs better than these baselines in terms of most MMD metrics. While the line Fiedler variant outperforms the DFS variant on the planar graph dataset w.r.t. VUN, the DFS variant performs substantially better on the SBM dataset. Notably, on the SBM dataset, the DFS variant outperforms all baselines w.r.t. VUN and many MMD metrics. On both datasets, ANFM appears competitive with the two diffusion-based approaches, DiGress and ESGG.

## 5.2 Experiments with Expanded Synthetic Datasets

We supplement the results presented above by training our model on larger synthetic datasets. Namely, we generate training sets consisting of 8192 graphs and corresponding validation and test sets consisting of 256 graphs each. We use the same data generation approach as Martinkus et al. (2022) to obtain expanded planar and SBM datasets. Additionally, we produce an expanded dataset of lobster graphs using NetworkX (Hagberg et al., 2008), as done in (Liao et al., 2019). We perform one training run per dataset for the DFS variant and three independent runs for the line Fiedler variant to illustrate robustness. We present the deviations observed across the three runs in Appendix F.2. We visualize samples from our model in Appendix F.3. In Table 2, we compare our method to our three baselines. For reasons of brevity, we only report the median performance here and refer to Appendix F.1 for a critical discussion of techniques to evaluate performance across multiple runs. All models reach perfect uniqueness and novelty scores on the expanded planar and SBM datasets and comparable uniqueness and novelty performance on the expanded lobster dataset (c.f. Appendix F.2). We find that ANFM is substantially faster during inference than the diffusion models, consistently achieving at least a 100-fold speedup in comparison to DiGress and ESGG. Moreover, we find that our method appears competitive with respect to sample quality, outperforming the two diffusion models on the expanded SBM dataset in terms of validity. On the expanded planar graph dataset, we observe that the line Fiedler ANFM variant outperforms or matches these baselines w.r.t. three of the four MMD metrics. For ESGG, we note that we obtain a surprisingly low validity score on the expanded SBM dataset and refer to Appendix G.5 for further discussion on this. We find that ANFM substantially outperforms the autoregressive baseline, GRAN, in terms of validity and MMD metrics.

Table 2: Performance on expanded synthetic datasets, evaluated on 1024 model samples. Showing a single run for the baselines and DFS, the median across three runs for the line Fiedler variant. *ESGG evaluation modified to draw graph sizes from empirical training distribution and use 100 refinement steps for determining SBM validity.

| | VUN (↑) | Deg. (↓) | Clus. (↓) | Orbit (↓) | Spec. (↓) | Time (↓) |
|---|---|---|---|---|---|---|
| | Expanded Planar Graphs ($|V| = 64$, $N_{\text{train}} = 8192$) | | | | | |
| GRAN | 0.19 | 0.0061 | 0.1862 | 0.0961 | 0.0081 | 0.0303 |
| DiGress | 80.76 | 0.0004 | 0.0217 | 0.0045 | 0.0024 | 2.73 |
| ESGG* | **89.94** | 0.0007 | **0.0162** | 0.0074 | **0.0012** | 4.65 |
| ANFM (Fdl.) | 79.20 | 0.0004 | 0.0183 | **0.0002** | **0.0012** | **0.0154** |
| ANFM (DFS) | 45.61 | **0.0003** | 0.0296 | 0.0004 | 0.0017 | 0.0164 |
| | Expanded SBM Graphs ($N_{\text{train}} = 8192$) | | | | | |
| GRAN | 25.29 | 0.0186 | 0.0086 | 0.0305 | 0.0022 | 0.133 |
| DiGress | 56.15 | **0.0002** | 0.0056 | 0.0076 | 0.0009 | 12.99 |
| ESGG* | 3.52 | 0.0949 | 0.0121 | 0.0518 | 0.0122 | 39.42 |
| ANFM (Fdl.) | 75.98 | 0.0014 | 0.0051 | 0.0180 | 0.0011 | **0.0396** |
| ANFM (DFS) | **78.03** | 0.0008 | **0.0049** | **0.0049** | **0.0008** | 0.0397 |
| | Expanded Lobster Graphs ($N_{\text{train}} = 8192$) | | | | | |
| GRAN | 41.99 | 0.0436 | 0.0069 | 0.1510 | 0.1469 | 0.0399 |
| DiGress | **96.58** | $9.79 \times 10^{-5}$ | $8.33 \times 10^{-7}$ | 0.0016 | 0.0009 | 4.86 |
| ESGG* | 63.96 | 0.0007 | **0.00** | 0.0027 | 0.0023 | 3.16 |
| ANFM (Fdl.) | 79.10 | 0.0004 | $7.89 \times 10^{-5}$ | 0.0010 | 0.0016 | 0.0175 |
| ANFM (DFS) | 87.60 | $\mathbf{7.82 \times 10^{-5}}$ | $1.64 \times 10^{-6}$ | **0.0007** | 0.0010 | **0.0160** |

Consistent with our previous experiments, we find that the DFS variant outperforms the line Fiedler variant on two of the three datasets in terms of the VUN metric. Hence, the exploration of more effective filtration strategies remains an important area for future investigation.

## 5.3 Experiments with Real-World Data

In this subsection, we present empirical results on the protein graph dataset introduced by Dobson & Doig (2003) and the GuacaMol (Brown et al., 2019) dataset consisting of drug-like molecules.

**Proteins.** While results have been reported for the protein dataset in previous works, we re-evaluate the baselines on 1024 model samples to reduce bias and variance in the reported metrics. We use a trained GRAN checkpoint provided by Liao et al. (2019) but re-train ESGG and DiGress, as no trained models are available. In Table 3, we find that our model is again substantially faster than the diffusion-based baselines. Moreover, it is also 10 times faster than GRAN while outperforming it with respect to all MMD metrics. Compared to diffusion-based models, the sample quality of our approach appears worse by most, but not all, MMD metrics. We note that the sub-quadratic asymptotic sampling complexity (w.r.t. the number of nodes) of ESGG starts to become noticeable at the large size of protein graphs, enabling substantially faster generation than DiGress. However, despite the cubic asymptotic complexity of ANFM, it remains over 100 times faster than ESGG.

**GuacaMol.** So far, we have focused on generating un-attributed graphs. However, molecular structures require node and edge labels to represent atom and bond types, respectively. Inspired by the approach of Li et al. (2020), we first train ANFM to generate un-attributed topologies and then assign node and edge

labels using a simple VAE. Details of this post-hoc labeling procedure can be found in Appendix A. We report established metrics for the GuacaMol benchmark: the Fréchet ChemNet distance (FCD) (Preuer et al., 2018) embeds the SMILES representations of valid generated molecules via a pre-trained language model and computes an optimal transport distance between the approximate generated and reference embedding distributions. The KL Divergence metric (Brown et al., 2019) determines various physicochemical parameters of the generated molecules and compares them to the reference distribution via the Kullback-Leibler divergence. Consistent with Brown et al. (2019), both metrics are reported as the exponential of the scaled negative FCD and KL divergence and should thus be maximized. In Table 4, we compare the performance of ANFM to several baselines from Vignac et al. (2023), namely an LSTM model trained on SMILES strings (Brown et al., 2019), graph MCTS (Brown et al., 2019), and NAGVAE (Kwon et al., 2020). Here, only DiGress and ANFM are general-purpose graph generation methods, while the other models are specific to molecule generation. Notably, ANFM demonstrates competitive performance with DiGress. Consistent with previous findings (Vignac et al., 2023; Xu et al., 2024; Qin et al., 2025), we observe that language models for SMILES strings (i.e., the LSTM model) remain strong baselines, outperforming general-purpose graph generators in terms of FCD and KL divergence.

Table 3: Performance of models on protein graph dataset. *Graph sizes are drawn from empirical training distribution.

| | Protein Graphs ($100 \leq |V| \leq 500$, $N_{\text{train}} = 587$) | | | | |
|---|---|---|---|---|---|
| | Deg. ($\downarrow$) | Clus. ($\downarrow$) | Orbit ($\downarrow$) | Spec. ($\downarrow$) | Time ($\downarrow$) |
| GRAN | 0.0025 | 0.0510 | 0.1539 | 0.0051 | 2.25 |
| DiGress | **0.0006** | 0.0234 | **0.0289** | 0.0014 | 72.27 |
| ESGG* | 0.0033 | **0.0216** | 0.0557 | **0.0008** | 19.48 |
| ANFM (Fdl.) | 0.0024 | 0.0464 | 0.0532 | 0.0024 | 0.194 |
| ANFM (DFS) | 0.0024 | 0.0370 | 0.0620 | 0.0020 | **0.191** |

Table 4: Performance of models on GuacaMol benchmark.

| | GuacaMol ($2 \leq |V| \leq 88$, $N_{\text{train}} \approx 1.3M$) | | | | |
|---|---|---|---|---|---|
| | Valid ($\uparrow$) | Unique ($\uparrow$) | Novel ($\uparrow$) | KL Div ($\uparrow$) | FCD ($\uparrow$) |
| LSTM | **95.9** | 100 | 91.2 | **99.1** | **91.3** |
| NAGVAE | 92.7 | 95.5 | 100 | 38.4 | 0.9 |
| MCTS | 100 | 100 | 99.4 | 52.2 | 1.5 |
| DiGress | 85.2 | 100 | **99.9** | 92.9 | 68.0 |
| ANFM (DFS) | 75.6 | 100.0 | 98.3 | 95.1 | 65.2 |

## 5.4 Ablation Studies

In this subsection, we demonstrate that teacher forcing, noise augmentation, and adversarial fine-tuning are crucial components of our method. The substantial performance gains from noise augmentation and adversarial fine-tuning, in particular, suggest that exposure bias is a significant factor affecting our autoregressive model. We also study the effect of filtration granularity, controlled by the hyper-parameter $T$. In Appendix H, we present extensive additional ablations, investigating alternative filtration functions, schedules, and node individualization schemes.

Table 5: Two ablation studies using the line Fiedler variant on the expanded planar graph dataset. Showing median $\pm$ maximum deviation across three runs. For the noise ablation, we train for 100k steps in stage I. For the finetuning ablation, we train for 200k steps in stage I.

| | Noise Ablation | | Finetuning Ablation | |
|---|---|---|---|---|
| | Stage I w/ Noise | Stage I w/o Noise | Stage II | Stage I w/ Noise |
| VUN ($\uparrow$) | **20.21** $_{\pm 3.22}$ | 0.00 $_{\pm 0.00}$ | **79.20** $_{\pm 7.13}$ | 23.24 $_{\pm 9.67}$ |
| Deg. ($\downarrow$) | **0.0058** $_{\pm 0.0008}$ | 0.0864 $_{\pm 0.0749}$ | **0.0004** $_{\pm 5.43 \times 10^{-5}}$ | 0.0036 $_{\pm 0.0009}$ |
| Clus. ($\downarrow$) | **0.1768** $_{\pm 0.0106}$ | 0.3179 $_{\pm 0.0037}$ | **0.0183** $_{\pm 0.0014}$ | 0.1547 $_{\pm 0.0280}$ |
| Spec. ($\downarrow$) | **0.0048** $_{\pm 0.0011}$ | 0.1042 $_{\pm 0.0760}$ | **0.0012** $_{\pm 0.0004}$ | 0.0033 $_{\pm 0.0014}$ |
| Orbit ($\downarrow$) | **0.0129** $_{\pm 0.0169}$ | 0.7115 $_{\pm 0.4411}$ | **0.0002** $_{\pm 0.0016}$ | 0.0043 $_{\pm 0.0023}$ |

**Noise Augmentation.** We find that noise augmentation of intermediate graphs substantially improves performance during training with teacher forcing (stage I). We illustrate this using the line Fiedler variant on the expanded planar graph dataset in Table 5 (left). A corresponding experiment for the DFS variant can be found in Appendix H.

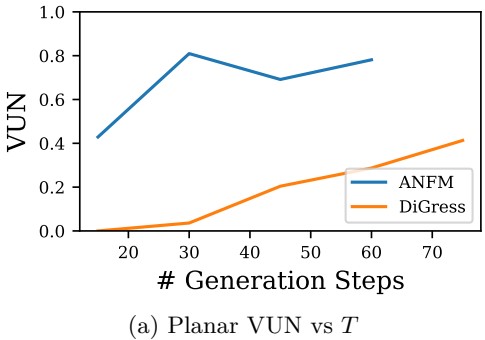
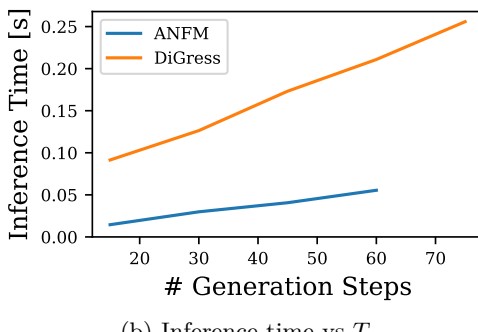

(a) Planar VUN vs $T$            (b) Inference time vs $T$

Figure 2: Performance of ANFM and DiGress on the expanded planar dataset as number of generation steps varies. The original (default) DiGress variant uses 1000 generation steps.

**GAN Tuning.** In Table 5 (right), we compare performance after training with teacher-forcing and subsequent adversarial fine-tuning on the expanded planar graph dataset. We find that adversarial fine-tuning substantially improves performance, both in terms of validity and MMD metrics. A corresponding analysis for the DFS variant and the expanded SBM and lobster datasets can be found in Appendix H.

**Teacher Forcing.** We illustrate in Table 6 the significance of the teacher forcing training stage (i.e. stage I). We compare the performance that ANFM achieves with adversarial fine-tuning using two different checkpoints collected during the first training stage. Initializing fine-tuning with an earlier checkpoint substantially harms performance, underscoring the important role of teacher forcing training in finding a near-optimal model before performing the adversarial fine-tuning.

Table 6: Performance of the line Fiedler ANFM variant on the expanded planar graph dataset with different training durations during stage I, followed by stage II training. Showing median across three runs for 200k steps and a single run for 10k steps.

| # Stage I Steps | VUN ($\uparrow$) | Deg. ($\downarrow$) | Clus. ($\downarrow$) | Orbit ($\downarrow$) | Spec. ($\downarrow$) |
|---|---|---|---|---|---|
| 200k | **79.20** | **0.0004** | **0.0183** | **0.0002** | **0.0012** |
| 10k | 3.32 | 0.0016 | 0.2278 | 0.0464 | 0.0069 |

**Filtration Granularity.** In Figure 2, we study the impact of filtration granularity, *i.e.,* the number of steps $T$, on generation quality and speed of ANFM. Analogously, we investigate how the number of denoising steps influences quality and speed in DiGress. We re-train the models with varying $T$. ANFM consistently outperforms DiGress in computational efficiency across all $T$. While DiGress only achieves a maximum VUN of 41% for our largest considered $T$, ANFM achieves a VUN of 81% for $T = 30$. We provide a complementary analysis of MMD metrics in Appendix L.

# 6 Conclusion

We proposed ANFM, an efficient autoregressive graph generative model based on graph filtration. ANFM generates high-quality graphs, outperforming existing autoregressive models and rivaling discrete diffusion approaches in terms of quality while being substantially faster at inference. Various ablations demonstrated the configurability of ANFM and indicated that exposure bias is an important challenge for autoregressive graph modeling. Our experiments suggested that ANFM's unique ability to construct non-montonic graph sequences and training strategies encouraging this behavior (noise augmentation and adversarial finetuning via RL) are the main drivers of performance.

We have argued that our proposed filtration approach generalizes existing autoregressive methods by allowing for sequences of subgraphs with non-induced edge sets. Although this affords additional flexibility

in constructing graph sequences for training, the non-induced sequences we investigated (line Fiedler filtrations) were often outperformed by sequences of subgraphs with induced edge sets (DFS filtrations). Thus, the exploration of alternative filtration strategies is left for future research.

While we have demonstrated that node and edge attributes may be sampled in a post-hoc fashion using a VAE, direct modeling within ANFM remains an interesting area for future investigation. We also highlight issues with existing benchmark datasets, where small sample sizes during evaluation lead to unreliable results with high variance and bias. Systematically addressing these challenges is essential for enabling more reliable and fair benchmarking in future research.

## Acknowledgements

This work was supported by the Max Planck Society. We thank Nina Corvelo Benz for her insightful feedback on the manuscript.

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

# Appendix

This appendix is organized as follows: We provide details on post-hoc attribute generation in Appendix A. In Section B, we analyze the runtime complexity of ANFM and investigate a variant that is asymptotically more efficient w.r.t. $T$. In Section C, we show that we optimize a lower-bound on the data log-likelihood in training stage I. Details on hyperparameters and model architecture are provided in Sections D and E. We provide evaluation results across several runs and qualitative model samples in Section F. Section G provides details on our baselines. Additional ablations on filtration function, schedule, node individualization, etc., are presented in Section H. We discuss the shortcomings of established evaluation approaches in Section I. Finally, we detail the adversarial finetuning algorithm in Section J.

## A  Generating Attributed Graphs

### A.1  Variational Inference

To generate attributed graphs $(G, l_V, l_E)$ with topology $G$ and node and edge labels $l_V$ and $l_E$, we propose to first generate the un-attributed graph topology using ANFM. Subsequently, we label nodes and edges using a separate model. I.e., we decompose the joint distribution as:

$$p_\theta(G, l_V, l_E) = p_\theta^{\text{label}}(l_E, l_V \mid G) \cdot p_\theta^{\text{ANFM}}(G) \tag{12}$$

where $p_\theta^{\text{label}}$ generates node and edge attributes. We take inspiration from Li et al. (2020) and train $p_\theta^{\text{label}}$ as a variational autoencoder.

**Variational Inference.**  Let $G := (V, E)$ be a graph topology from the training dataset and let $l_V : V \to \mathcal{X}$ and $l_E : E \to \mathcal{Y}$ be ground-truth node and edge labels for $G$, respectively. An encoder $q_\phi$ maps the labeled graph $(G, l_V, l_E)$ to a distribution over $D$-dimensional latent node representations $\boldsymbol{z} \in \mathbb{R}^{V \times D}$. A decoder $p_\theta$ maps a tuple $(G, \boldsymbol{z})$, i.e. the graph topology combined with a latent representation, to a distribution over node and edge attributes. In practice, we consider discrete attributes and $p_\theta$ parametrizes a product distribution across nodes and edges. If $\mathcal{X}$ or $\mathcal{Y}$ factorize into a product of simpler spaces (i.e., when there are multiple node or edge attributes), $p_\theta$ parametrizes a corresponding product distribution over $\mathcal{X}$ or $\mathcal{Y}$. The encoder $q_\phi$ parametrizes a Gaussian distribution with a diagonal covariance matrix. We formulate the typical evidence lower bound (Kingma & Welling, 2014):

$$p_\theta(l_E, l_V \mid G) \geq \mathbb{E}_{\boldsymbol{z} \sim q_\phi(\,\cdot\,\mid G, l_E, l_V)} \left[ p_\theta(l_E, l_V \mid G, \boldsymbol{z}) \right] - D_{\text{KL}} \left( q_\phi(\,\cdot\,\mid G, l_E, l_V) \mid \mathcal{N}(\boldsymbol{0}, I) \right) \tag{13}$$

**Architecture.**  We implement $q_\phi$ and $p_\theta$ as GraphGPS (Rampásek et al., 2022) models. Since $q_\phi$ operates on edge-labeled graphs, we use GINE message passing layers (Hu et al., 2020) within its graph transformer. The decoder $p_\theta$, on the other hand, only operates on node representations, and we therefore use GIN layers (Xu et al., 2019). The two GNNs are provided with Laplacian and random walk positional embeddings (Dwivedi et al., 2022) and we use 5% dropout layers (Srivastava et al., 2014). We feed the node representations produced by $p_\theta$ through MLPs to generate distributions over node attributes. To generate a distribution over the attribute of an edge $\{u, v\}$, we add the node representations corresponding to $u$ and $v$ and feed the resulting vector through an MLP.

**Training.**  We train $q_\phi$ and $p_\theta$ jointly by maximizing a re-weighted version of the ELBO in Eqn. (13). Specifically, we divide the loss for a given datum $(G, l_E, l_V)$ by $|V|$, the cardinality of the node set of $G$.

**Sampling.**  To sample attributes, we draw latent node representations $\boldsymbol{z}$ from the standard normal distribution. We select the maximum likelihood node and edge attributes from $p_\theta(\,\cdot\,\mid G, \boldsymbol{z})$. The encoder $q_\phi$ may be discarded after training.

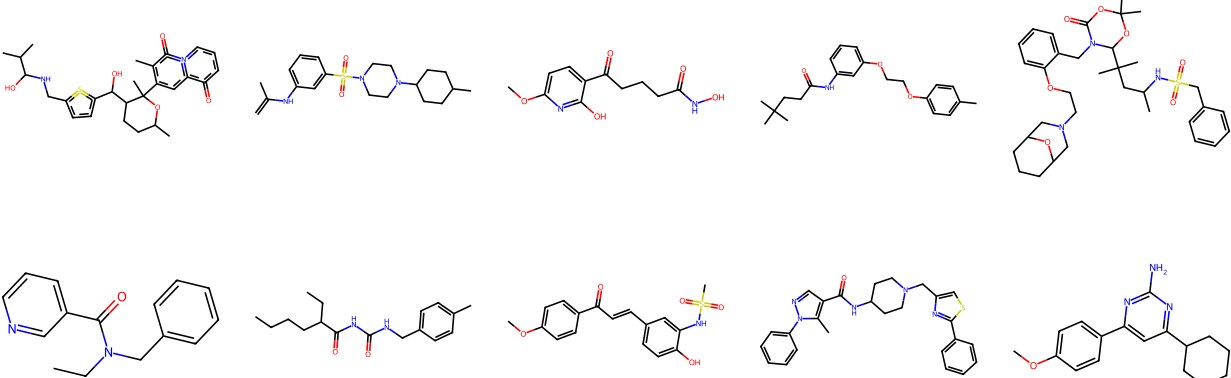

Figure 3: Uncurated samples from ANFM model trained on GuacaMol.

## A.2 Details on GuacaMol Experiments

Below, we provide details on the molecule graph generation presented in Sec. 5.3. As described in Appendix A, we train ANFM on the un-attributed molecule topologies of GuacaMol. Independently, we train a VAE to generate attributes conditioned on a topology. To sample a molecule, we first generate a graph using ANFM and label nodes and edges using the VAE.

**Attributes.** We produce edge labels that indicate the bond type between heavy atoms, distinguishing between single, double, triple, and aromatic bonds. To reconstruct the correct molecule from its graph representation, we find that we require require several node attributes. Namely, given an atom (i.e., node), we generate its type (i.e., the element), the number of explicit hydrogens bound to it, the number of radical electrons, and its partial charge.

**ANFM Hyperparameters.** We train an ANFM model using the DFS filtration strategy. In stage I, we use similar hyper-parameters as for the other datasets. We use $T = 30$ filtration steps, a learning rate of $10^{-5}$, a local batch size of 64 on two GPUs, and 300k training steps. Throughout stage I training, we monitor validation loss and ensure that the model does not overfit. During training stage II, we use a batch size of 32 and a learning rate of $2.5 \times 10^{-8}$ for the generative model. The discriminator has 3 layers and a hidden dimension of 32. The other hyperparameters of the discriminator and value model match those used in the other experiments.

**VAE Hyperparameters.** We use 5-layer GraphGPS (Rampásek et al., 2022) models for the encoder and decoder, respectively. The hidden dimension is 256 while we choose the latent dimension $D$ to be 8. The Laplacian positional embedding is 3-dimensional while the random walk positional embedding is 8-dimensional. We train for 250 epochs on the GuacaMol dataset, using a batch size of 512, an Adam optimizer (Kingma & Ba, 2015) and a learning rate of $10^{-4}$.

**Model Samples.** In Figure 3, we show uncurated samples from the ANFM model.

# B Sampling Complexity of ANFM

## B.1 Complexity Analysis

In the following, we analyze the asymptotic runtime complexity of sampling a graph from our proposed model and the baselines we studied in Section 5.

**Proposition B.1.** *The asymptotic runtime complexity for sampling a graph with $N$ nodes from an ANFM with $T$ timesteps is:*

$$\mathcal{O}(T^2 N + T N^3) \tag{14}$$

*Proof.* To sample a graph from an ANFM, one has to perform $T$ forward passes through our proposed mixer architecture. These forward passes are preceded by the computation of various graph features, including Laplacian eigenvalues and eigenvectors. This eigendecomposition has complexity $\mathcal{O}(N^3)$. At timestep $0 \leq t < N$, the structural mixing layers have complexity $\mathcal{O}(N^2)$ due to the self-attention component of SAT. The temporal mixing layers, on the other hand, have complexity $\mathcal{O}(N(t+1))$, as each node attends to its representations at timesteps $0, \ldots, t$. We bound this complexity by $\mathcal{O}(NT)$. Hence, aggregating these complexities across all $T$ timesteps, we obtain the following runtime complexity:

$$\mathcal{O}(T^2 N + T N^2 + T N^3) = \mathcal{O}(T^2 N + T N^3) \tag{15}$$

$\square$

Below we show that the asymptotic complexity of ANFM differs from the complexity of DiGress only in the quadratic term w.r.t. $T$:

**Proposition B.2.** *The asymptotic runtime complexity for sampling a graph with $N$ nodes from a DiGress model with $T$ denoising steps is:*

$$\Omega(T N^3) \tag{16}$$

*Proof.* Similar to ANFM, DiGress performs an eigendecomposition of the graph laplacian in each denoising step. Hence, one obtains a complexity of $\Omega(N^3)$ in each timestep, resulting in an overall complexity of $\Omega(T N^3)$. $\square$

We further analyze the asymptotic complexities of our other baselines of Sec. 5.2 and Sec. 5.3.

**Proposition B.3.** *The asymptotic runtime complexity for sampling a graph with $N$ nodes from a GRAN model is $\Omega(N^2)$.*

*Proof.* GRAN explicitly constructs a dense adjacency matrix with $N^2$ entries. $\square$

**Proposition B.4.** *The asymptotic runtime complexity of sampling a graph with $N$ nodes and $M$ edges from an ESGG model is $\Omega(N + M)$.*

*Proof.* This bound should trivially be satisfied by any generative model, as one already needs $\Omega(N + M)$ bits to represent a graph with $M$ edges on $N$ nodes. We refer to (Bergmeister et al., 2024) for a discussion on how tight this bound is. $\square$

In Table 7, we summarize these asymptotic complexities. While this analysis may suggest that ANFM

Table 7: Asymptotic runtime complexities for sampling from different graph generative models.

| Method | Sampling complexity |
|---------|---------------------|
| ANFM | $\mathcal{O}(T^2 N + T N^3)$ |
| DiGress | $\Omega(T N^3)$ |
| GRAN | $\Omega(N^2)$ |
| ESGG | $\Omega(N + M)$ |

does not scale well to extremely large graphs, we caution the reader that the asymptotic behavior may not accurately reflect efficiency in practice: Firstly, multiplicative constants and lower-order terms are ignored. Hence, it remains unknown in which regimes the asymptotic behavior governs inference time. Secondly, the analysis was made under the assumption that hyper-parameter choices (i.e. depth, width, etc.) is kept

constant as $N$ and $M$ increase. It is reasonable to expect that more expressive networks are required to model large graphs.

## B.2 A First-Order Autoregressive Variant

As we demonstrated in Appendix B.1, the runtime of ANFM is quadratic in the number of generation steps $T$ due to the temporal mixing operations which are implemented as transformer decoder layers. Analogously, one may verify that the memory complexity of sampling from ANFM is linear in $T$. In this subsection, we study a simplified variant of ANFM in which we use a first-order autoregressive structure. I.e., we enforce:

$$p_\theta(\tilde{G}_{t+1}|\tilde{G}_t, \ldots, \tilde{G}_0) = p_\theta(\tilde{G}_{t+1}|\tilde{G}_t) \tag{17}$$

We implement this by ablating the causally masked self-attention mechanism from the transformer layers in our mixer model, leaving only the feed-forward modules. The resulting first-order variant of ANFM has space complexity which is independent of $T$ and runtime complexity which is linear in $T$.

We train such a first-order variant of ANFM on the expanded planar graph dataset, using the line Fiedler filtration strategy and the same hyperparameters as for the transformer-based variant (see Appendix D.2). Using the first-order variant, we observe training instabilities after the first 100k training steps of stage I. While reducing the learning rate rectifies this instability, we find that this slows learning progress substantially. Instead, we use a model checkpoint at 100k steps and continue with training stage II.

In Table 8, we compare the performance of the transformer-based and the first-order variants after 100k steps of stage I training. In Table 9, we compare the performance after the subsequent stage II training. While we perform only 100k training steps in stage I for the first-order variant, we perform 200k training steps for the transformer-based variant, as it did not exhibit instabilities.

Table 8: Performance of two ANFM variants on the expanded planar graph dataset after 100k steps of stage I training. Showing median across three runs for the transformer-based variant and a single run for the first-order variant. All models reach perfect uniqueness and novelty scores.

|  | VUN ($\uparrow$) | Deg. ($\downarrow$) | Clus. ($\downarrow$) | Orbit ($\downarrow$) | Spec. ($\downarrow$) |
|---|---|---|---|---|---|
| Transformer | 20.21 | 0.0058 | 0.1768 | 0.0129 | 0.0048 |
| First-Order | 5.66 | 0.0004 | 0.1782 | 0.0041 | 0.0035 |

Table 9: Performance of two ANFM variants on the expanded planar graph dataset after stage II training. The transformer-based variant was trained for 200k steps in stage I while the first-order variant was trained for 100k steps in stage I. Showing median across three runs for the transformer-based variant and a single run for the first-order variant. All models reach perfect uniqueness and novelty scores.

|  | VUN ($\uparrow$) | Deg. ($\downarrow$) | Clus. ($\downarrow$) | Orbit ($\downarrow$) | Spec. ($\downarrow$) | Time ($\downarrow$) |
|---|---|---|---|---|---|---|
| Transformer | 79.20 | 0.0004 | 0.0183 | 0.0002 | 0.0012 | 0.0278 |
| First-Order | 70.02 | 0.0004 | 0.0229 | 0.0046 | 0.0013 | 0.0247 |

Generally, we observe that the transformer-based ANFM variant slightly outperforms the first-order variant in terms of quality. However, the first-order variant remains competitive after stage II trainig and, thus, may be a suitable alternative in cases where a large $T$ is chosen. In our setting ($T = 30$), however, we find that the first-order variant is not substantially faster during inference, indicating that the runtime is not governed by the quadratic complexity in $T$.

# C   A Bound on Model Evidence in ANFM

Given a graph $G_T$, let $q(G_{T-1}, \ldots G_1 | G_T)$ be the data distribution over noisy filtrations of this graph, determined by the choice of filtration function, scheduling, and noise augmentation. We assume that $G_0$ is deterministically the completely disconnected graph. Moreover, we note that by applying our noise augmentation strategy we ensure that $q$ is supported everywhere. Given some graph $G_T$, we can now derive the following evidence lower bound:

$$
\begin{aligned}
\log p_\theta(G_T) =\ & \log \sum_{G_1, \ldots, G_{T-1} \in \mathcal{G}} p_\theta(G_T, \ldots, G_0) \\
=\ & \log \sum_{G_1, \ldots, G_{T-1} \in \mathcal{G}} q(G_{T-1}, \ldots, G_1 | G_T) \frac{p_\theta(G_T, \ldots, G_0)}{q(G_{T-1}, \ldots, G_1 | G_T)} \\
\geq\ & \mathbb{E}_{(G_{T-1}, \ldots, G_1) \sim q(\,\cdot\,| G_T)} \left[ \log \frac{p_\theta(G_T, \ldots, G_0)}{q(G_{T-1}, \ldots, G_1 | G_T)} \right] \\
=\ & \mathbb{E}_{(G_{T-1}, \ldots, G_1) \sim q(\,\cdot\,| G_T)} \left[ \sum_{t=1}^{T} \log p_\theta(G_t | G_{t-1}, \ldots, G_0) - \log q(G_{T-1}, \ldots G_1 | G_T) \right]
\end{aligned}
\tag{18}
$$

We note that this lower bound is (up to sign and a constant that does not depend on $\theta$) exactly the autoregressive loss we use in training stage I. Hence, while we train ANFM to model sequences of graphs, we do actually optimize an evidence lower bound for the final graph samples $G_T$.

# D   Hyperparameters

## D.1   Practical Advice on Hyperparameter Choice

In the following, we provide some practical advice on choosing some of the most important hyper-parameters in ANFM. Generally, we tuned few hyper-parameters in our experiments. We found the number of generation steps $T$ to be one of the most impactful hyper-parameters.

**Filtration Function.**   The filtration function $f : E \to \mathbb{R}$ is the main component determining the structure of the graph sequence during stage I training. We recommend that $f$ should convey meaningful information about the structure and assign (mostly) distinct values to distinct edges. We note that if $f$ fails to assign distinct values to edges, many edges may be added in a single generation step, regardless of the choice of $T$. We found both the edge Fiedler function and the DFS filtrations to perform well in many settings. We recommend that practitioners utilize these filtration strategies and perform experiments with further filtration functions that incorporate domain-specific inductive biases. In the case of generating protein graphs, for instance, one may consider a filtration function that quantifies the distance of residues in the sequence (this filtration would first generate a backbone path, followed by increasingly long-range interactions of residues).

**Filtration Granularity.**   As we demonstrate in Sec. 5.4, the choice of the number of generation steps $T$ has a substantial influence on sampling efficiency and generation quality. Generally, $T$ can be chosen substantially smaller than in other autoregressive models. In our experiments, we chose $T \leq 32$. We recommend that practitioners experiment with different values in this order of magnitude. We further caution that increasing $T$ does not necessarily improve sample quality and may actually harm it.

**Scheduling Function.**   The filtration schedule governs the rate at which edges are added at different timesteps. In the case of the line Fiedler filtration function, it is determined by $\gamma$, which should be monotonically increasing with $\gamma(0) = 0$ and $\gamma(1) = 1$. We found the heuristic choice of $\gamma(t) := t$ to work well in many settings. However, as we demonstrate in Appendix H, the concave schedule may be a promising alternative. We recommend that practitioners validate stage I training with a convex, linear, and concave scheduling function. We note that the scheduling function is no longer used during stage II training, as the

model is left free to generate arbitrary intermediate graphs. Hence, performance after stage I training may be a suitable metric for selecting a scheduling function.

**Noise Augmentation.** We use noise augmentation during training stage I to counteract exposure bias, i.e. the accumulation of errors in the sampling trajectory. Manual inspection of the graph sequence $\tilde{G}_0, \ldots, \tilde{G}_T$ may be difficult. However, we found that inspecting the development of the edge density over this graph sequence can provide a simple tool for diagnosing exposure bias. In models that do not utilize noise augmentation, we can observe that, after some generation steps, the edge density oftentimes deviates from its expected behavior (e.g. by suddenly increasing or dropping substantially). In this case we expect that noise augmentation can rectify exposure bias. In our experiments, we find that we do not need to tune the noise schedule. Instead, we fix a single schedule that is shared across all models. For details on this schedule, we refer to Appendix D.2.

**Perturbation of Node Orderings.** During training stage I, ANFM variants using the line Fiedler filtration function may overfit on small datasets. This manifests as an increase in validation loss, while the validation MMD metrics continue to improve. We observe this behavior only on the small datasets in Sec. 5.1 and find that it can be mostly attributed to the node ordering used to derive initial node representations (c.f. Appendix E.1). We recommend to monitor validation losses during stage I training. If the validation loss starts to slowly increase while the training loss continues to decrease, we recommend to randomly perturb the node ordering, as described in Appendix E.1. The noise scale $\sigma$ should be increased until no over-fitting can be observed. We find that the DFS filtration strategy is less prone to overfitting, as DFS node orderings are not unique. Hence, we may include many distinct filtrations of a single graph $G_T$ in our training set.

### D.2 ANFM Hyperparameters

In Tables 10 and 11, we summarize the most important hyperparameters of the generative model used in our experiments, including the number of filtration steps ($T$), mixture components ($K$), learning rate (LR), batch size (BS) in the format `num_gpus` $\times$ `grad_accumulation` $\times$ `local_bs`, and the number of perturbed filtration sequences we produce per graph in our training set (# Perturbations).

Table 10: Hyper-parameters for line Fiedler variant of ANFM.

| | SPECTRE Planar | SPECTRE SBM | Expanded Planar | Expanded SBM | Expanded Lobster | Protein |
|---|---|---|---|---|---|---|
| $T$ | 30 | 15 | 30 | 15 | 30 | 15 |
| $K$ | 8 | 4 | 8 | 4 | 8 | 16 |
| # Layers | | | 5 | | | |
| Hidden Dim | | | 256 | | | |
| Laplacian PE dim. | | | 4 | | | |
| RWPE dim. | | | 20 | | | |
| Noise Augm. | | $\lambda_t$ affine with $\lambda_1 = 0.25$ and $\lambda_{T-1} = 0.05$ | | | | |
| # Perturbations | 256 | 256 | 4 | 4 | 4 | 8 |
| Perturb Node Order | Yes | Yes | No | No | No | No |
| Stg. I LR | $2.5 \times 10^{-5}$ | $1 \times 10^{-5}$ | $2.5 \times 10^{-5}$ | $1 \times 10^{-5}$ | $1 \times 10^{-5}$ | $1 \times 10^{-5}$ |
| Stg. I $\ell^2$ grad. clip | | | None | | | |
| Stg. I BS | $2 \times 1 \times 32$ | $2 \times 1 \times 32$ | $2 \times 1 \times 32$ | $2 \times 1 \times 32$ | $2 \times 1 \times 32$ | $2 \times 4 \times 8$ |
| # Stg. I Steps | 50k | 100k | 200k | 200k | 100k | 100k |
| Stg. I Precision | | | BF16 AMP | | | |
| Stg. II LR | | | $1.25 \times 10^{-7}$ | | | |
| Stg. II BS | $1 \times 4 \times 32$ | $1 \times 4 \times 32$ | $1 \times 4 \times 32$ | $1 \times 4 \times 32$ | $1 \times 4 \times 32$ | $1 \times 16 \times 8$ |
| # Stg. II Iters | 2.5k | 3k | 1.5k | 5k | 4k | 1.5k |

In Table 12, we additionally provide the most important hyperparameters of the discriminator and value model trained during the adversarial fine-tuning stage.

Table 11: Hyper-parameters for DFS variant of ANFM. Parameters that do not appear in this table exactly match the corresponding parameters for the line Fiedler variant, c.f. Table 10.

| | SPECTRE Planar | SPECTRE SBM | Expanded Planar | Expanded SBM | Expanded Lobster | Protein |
|---|---|---|---|---|---|---|
| $T$ | 32 | 15 | 32 | 15 | 30 | 15 |
| # Perturbations | 1,024 | 512 | 32 | 16 | 32 | 16 |
| Perturb Node Order | | | No | | | |
| Stg. I LR | $1 \times 10^{-4}$ | $5 \times 10^{-5}$ | $1 \times 10^{-4}$ | $5 \times 10^{-5}$ | $5 \times 10^{-5}$ | $1 \times 10^{-5}$ |
| Stg. I $\ell^2$ grad. clip | 75 | 250 | 75 | 250 | 75 | 75 |
| # Stg. I Steps | 200k | 200k | 200k | 200k | 100k | 200k |
| # Stg. II Iters | 1k | 1k | 1.5k | 5k | 4k | 530 |

Table 12: Hyper-parameters of discriminator and value model used during adversarial fine-tuning.

| | | SPECTRE Planar | SPECTRE SBM | Expanded Planar | Expanded SBM | Expanded Lobster | Protein |
|---|---|---|---|---|---|---|---|
| Disc. | LR | | | $1.00 \times 10^{-4}$ | | | |
| | BS | | | $1 \times 1 \times 32$ | | | |
| | # Layers | 2 | 3 | 3 | 3 | 3 | 2 |
| | Hidden dim. | 32 | 128 | 128 | 128 | 128 | 64 |
| | RWPE dim. | 5 | 20 | 20 | 20 | 20 | 20 |
| Val. | LR | | | $2.50 \times 10^{-4}$ | | | |
| | BS | | | $1 \times 4 \times 32$ | | | |
| | # Layers | | | 5 | | | |
| | Hidden dim. | | | 128 | | | |

# E    Details on Architecture

## E.1    Input Node Representations

We define the input node representations as:

$$v_i^{(t)} := f_\theta(G_t)_i + W_i^{\mathrm{node}}, \tag{19}$$

where $f_\theta$ produces node features from Laplacian positional encodings (Dwivedi et al., 2023), random walk encodings (Dwivedi et al., 2022), and cycle counts following DiGress (Vignac et al., 2023). For precise definitions of these features, we refer to Appendix E.2. The matrix $W^{\mathrm{node}} \in \mathbb{R}^{N \times D}$ is a trainable embedding layer where $N$ denotes the cardinality of the largest vertex set seen during training. It is important to note that the computation of input node representations requires a specific node ordering to index $W^{\mathrm{node}}$. While the permutation equivariance of our model and the symmetry of the initially empty graph $G_0$ allow for arbitrary ordering during inference, we employ a structured approach during teacher-forcing training. This ordering is derived from the structure of the graph $G_T$ and depends on the filtration strategy.

**Node Ordering for DFS Variant.**    The filtration function of the DFS variant is derived from a DFS node ordering of $G_T$. Hence, we simply use this node ordering to assign positional embeddings.

**Node Ordering for Line Fiedler Variant.**    For the line Fiedler variant, we propose a node weighting scheme $h : V \to \mathbb{R}$ defined as:

$$h(i) := \frac{1}{|\mathcal{N}_G(i)|} \sum_{j \in \mathcal{N}_G(i)} f(e_{ij}), \qquad \forall \, i \in V, \tag{20}$$

where $\mathcal{N}_G(i)$ represents the neighborhood of node $i$ in $G$. This weighting assigns to each node the average weight of its incident edges, as determined by the filtration function $f$. We then establish a node ordering such that $h$ is non-increasing. The impact of different ordering strategies on model performance is further studied and compared in Appendix H.

When training on small datasets, such as those introduced by Martinkus et al. (2022), we find that the node individualization in Eqn. (19) can lead to overfitting. This manifests as an increase in validation loss, while

the evaluation metrics (i.e. MMD and VUN) continue to improve. As a data augmentation strategy to avoid overfitting, we propose to add Gaussian noise to the node weights $h_G$ defined in Eqn. (19) when training on small datasets. I.e., we use the perturbed node weights

$$h_G(s) + \mathcal{N}(0, \sigma^2) \tag{21}$$

for sorting the nodes. We emphasize that this measure is independent of the perturbation of intermediate graphs introduced in Sec. 4.2. Moreover, we perturb node orderings only in the experiments on the small SPECTRE datasets (i.e., in Sec 5.1).

## E.2 Positional and Structural Encodings

In this section, we describe the positional and structural encodings (PSE) we use to construct $f_\theta(G_t)$. The positional and structural encodings are obtained by stacking Laplacian positional embeddings, random walk encodings, and cycle counts: $\text{PSE} := \text{LPE} \parallel \text{RWE} \parallel \text{CC} \in \mathbb{R}^{N \times d}$. In the following, we describe how each of the three components is computed. Let $A$ be the adjacency matrix of $G_t$ and let $D \in \mathbb{R}^{N \times N}$ be the diagonal matrix containing the node degrees.

**Laplacian Positional Encoding.** For $k > 0$, the $k$-dimensional Laplacian positional encoding is defined as the first $k$ orthonormal eigenvectors with non-zero eigenvalues of the symmetrically normalized Laplacian matrix $L \in \mathbb{R}^{N \times N}$:

$$L_{i,j} := \begin{cases} 1, & \text{if } i = j \text{ and } d_i \neq 0, \\ -\dfrac{1}{\sqrt{D_{ii}D_{jj}}}, & \text{if } i \neq j \text{ and } A_{ij} = 1, \\ 0, & \text{otherwise.} \end{cases} \tag{22}$$

These eigenvectors are stacked into the matrix $\text{LPE}_1 \in \mathbb{R}^{N \times k}$. In addition, we compute the corresponding eigenvalues $v \in \mathbb{R}^k$ and replicate them across nodes: $\text{LPE}_2 := \mathbb{1}_N v^\top \in \mathbb{R}^{N \times k}$. The final Laplacian positional encoding is obtained by stacking eigenvectors and replicated eigenvalues: $\text{LPE} := \text{LPE}_1 \parallel \text{LPE}_2 \in \mathbb{R}^{N \times 2k}$.

**Random Walk Encoding.** For $k > 0$, the $k$-dimensional random walk structural encoding of vertex $i \in \{1, \ldots, N\}$ is defined as the vector

$$\begin{bmatrix} \text{RW}_{ii} & \text{RW}_{ii}^2 & \ldots & \text{RW}_{ii}^k \end{bmatrix}^\top \in \mathbb{R}^k \tag{23}$$

where $\text{RW} := AD^{-1} \in \mathbb{R}^{N \times N}$ is the random walk operator. The random walk encodings of all nodes form the random walk encoding $\text{RWE} \in \mathbb{R}^{N \times k}$.

**Cycle Counts.** For each vertex, we count the number of distinct 3- and 4-cycles the vertex participates in. Additionally, we count the total number of 3-, 4-, 5-, and 6-cycles contained in the graph. These counts are computed via closed-form formulas from Vignac et al. (2023). We refer to the published code and Appendix B2 of Vignac et al. (2023) for the exact formulas. Graph-level counts are replicated across nodes analogous to the Laplacian eigenvalues and stacked with node-level counts, producing $\text{CC} \in \mathbb{R}^{N \times 5}$.

## E.3 Edge Decoder Architecture

In this subsection, we present details on the edge decoder $D_{\cdot,\theta}$. While our approach is in principle applicable to discretely labeled edges, we concentrate on predicting distributions over unlabeled edges here. Fix some timestep $0 \leq t < T$. Assume that for this timestep, we are given some node representations $(v_i)_{i=1}^{|V|}$ produced by the backbone model. The edge decoder contains $K$ submodules that produce multivariate Bernoulli distributions. Assuming that the node-representations produced by the backbone are $D$-dimensional, let $\text{Dense}_{k,\theta}^{(1)} : \mathbb{R}^D \to \mathbb{R}^{2D}$ and $\text{Dense}_{k,\theta}^{(2)}, \text{Dense}_{k,\theta}^{(3)} : \mathbb{R}^{2D} \to \mathbb{R}^{2D}$ be fully connected layers learned for each component $k$. Define corresponding MLPs:

$$\text{MLP}_{k,\theta} := \text{ReLU} \circ \text{Dense}_{k,\theta}^{(2)} \circ \text{ReLU} \circ \text{Dense}_{k,\theta}^{(1)} \tag{24}$$

For each $k$, we process the node representations $v_i$ separately and split the resulting vectors into two $D$-dimensional halves:

$$(x_i^{(k)}, y_i^{(k)}) := \text{MLP}_{k,\theta}(v_i) \qquad (\hat{x}_i^{(k)}, \hat{y}_i^{(k)}) := \text{Dense}_k^{(3)}\left((x_i^{(k)}, y_i^{(k)})\right) \tag{25}$$

We define the logit $l_{k,i,j}$ for the presence of an edge and the logit $r_{k,i,j}$ for the absence of an edge:

$$l_{k,i,j} := \frac{x_i^{(k)\top}\hat{x}_j^{(k)} + x_j^{(k)\top}\hat{x}_i^{(k)}}{2} \qquad r_{k,i,j} := \frac{y_i^{(k)\top}\hat{y}_j^{(k)} + y_j^{(k)\top}\hat{y}_i^{(k)}}{2} \tag{26}$$

Finally, we define the likelihood of the presence of an edge as:

$$D_{k,\theta}(v_i, v_j) := \frac{\exp(l_{k,i,j})}{\exp(l_{k,i,j}) + \exp(r_{k,i,j})} \tag{27}$$

While this modeling of the mixture distributions is quite involved, it allows the edge decoder to be easily extended to produce distributions over labeled edges by producing logits for labels of node pairs (instead of producing logits for presence and absence of edges).

Finally, we compute a mixture distribution $\pi \in \Delta^{K-1}$ via $D_{\text{mix},\theta}$. To this end, we learn a node-level MLP:

$$\text{MLP}_{\text{mix},\theta}^{(1)} := \text{ReLU} \circ \text{Dense}_{\text{mix},\theta}^{(1)} \tag{28}$$

and a graph-level MLP:

$$\text{MLP}_{\text{mix},\theta}^{(2)} := \text{Dense}_{\text{mix},\theta}^{(3)} \circ \text{ReLU} \circ \text{Dense}_{\text{mix},\theta}^{(2)} \tag{29}$$

where $\text{Dense}_{\text{mix},\theta}^{(3)} : \mathbb{R}^D \to \mathbb{R}^K$. We then define:

$$D_{\text{mix},\theta}\left((v_i)_{i=1}^{|V|}\right) := \text{softmax}\left(\text{MLP}_{\text{mix},\theta}^{(2)}\left(\frac{1}{|V|}\sum_{i=1}^{|V|}\text{MLP}_{\text{mix},\theta}^{(1)}(v_i)\right)\right) \tag{30}$$

In the following, we discuss how our approach, and the edge decoder in particular, may be extended to edge-attributed and directed graphs.

**Edge Attributes.** While we only present experiments on un-attributed graphs, we note that our approach (in particular the edge decoder) is naturally extendable to discretely edge-attributed graphs. Assuming that one has $S$ possible edge labels (where one edge label encodes the absence of an edge), one would predict $S$ logits $l_{k,i,j}^{(s)}$ instead of predicting only two logits $l_{k,i,j}$ and $r_{k,i,j}$. Then, for fixed $i, j, k$, the vector

$$\text{softmax}_s\left(l_{k,i,j}^{(s)}\right)_{s=1}^{S} \qquad \in \qquad \Delta^{S-1} \tag{31}$$

would provide a distribution over labels for edge $\{v_i, v_j\}$. This distribution would be incorporated into a mixture (over $k$) of categorical distributions as above. Eqn. (10) would be adjusted to quantify the likelihood of edge labels instead of the likelihood of edge presence/absence.

**Directed Graphs.** In our experiments, we only consider applications of ANFM to undirected graphs. However, our approach is also naturally extendable to directed graphs. Concretely, one would first adjust all GNNs in ANFM to take edge directionality into account. One would additionally modify the product in Eqn. (10) to run over the entire adjacency matrix instead of only considering the upper triangle. I.e., one would get:

$$p_\theta(E_t|G_{t-1},\dots,G_0) := \sum_{k=1}^{K}\pi_k\prod_{i,j}\left\{\begin{array}{ll}p_k^{(i,j)} & \text{if} \quad e_{ij} \in E_t \\ 1 - p_k^{(i,j)} & \text{else}\end{array}\right\}. \tag{32}$$

Finally, the edge decoder would be adjusted in Eqn. (26) to drop the symmetrization of $l_{k,i,j}$ and $r_{k,i,j}$ w.r.t. $i$ and $j$ (i.e., one no longer enforces the presence of the edge $(v_i, v_j)$ to have the same probability as the presence of $(v_j, v_i)$).

## F    Extended Evaluation Results

### F.1    Evaluation Methodology for Analyzing Performance Across Multiple Runs

While previous works in graph generation oftentimes present results for only a single training run per dataset, we believe that it is important to provide some evidence that our proposed method yields stable results across different runs. We provide this evidence by performing three independent training runs with the line Fiedler filtration strategy on the expanded synthetic datasets and reporting median performance in Section 5.2 and the maximum deviation across the three runs in Appendix F.2.

Concretely, we perform three training runs and compute corresponding performance metrics $x_1 \leq x_2 \leq x_3$. We report the median as $m := x_2$ and the maximum deviation as $d := \max(|x_1 - m|, |x_3 - m|)$.

We settle on this evaluation methodology for the following practical reasons:

- Since training is expensive, we can only perform three training runs per dataset. These are too few trials to reliably determine empirical standard deviations.

- The median and maximum deviation allow straightforward reasoning about the outcomes of all three runs: $m - d$ lower-bounds the performance of the worst run, one run achieved exactly the median performance $m$, and the best run achieved at least the median performance, potentially exceeding it.

### F.2    Comprehensive Evaluation Results on Expanded Synthetic Datasets

In Table 13, we present the deviations observed across the three training runs discussed in Sec. 5.2. Additionally, in Table 14, we present uniqueness and novelty metrics for ANFM and our baselines.

Table 13: Full evaluation results for the Fiedler variant of ANFM trained on expanded synthetic datasets. Showing median across three runs $\pm$ maximum deviation.

|  | Expaned Planar | Expanded SBM | Expanded Lobster |
|---|---|---|---|
| VUN ($\uparrow$) | $79.20 \pm 7.13$ | $75.98 \pm 3.71$ | $79.10 \pm 7.13$ |
| Degree ($\downarrow$) | $0.0004 \pm 5.43 \times 10^{-5}$ | $0.0014 \pm 0.0062$ | $0.0004 \pm 0.0013$ |
| Clustering ($\downarrow$) | $0.0183 \pm 0.0014$ | $0.0051 \pm 0.0009$ | $7.89 \times 10^{-5} \pm 5.32 \times 10^{-5}$ |
| Spectral ($\downarrow$) | $0.0012 \pm 0.0004$ | $0.0011 \pm 0.0006$ | $0.0016 \pm 0.0028$ |
| Orbit ($\downarrow$) | $0.0002 \pm 0.0016$ | $0.0180 \pm 0.0171$ | $0.0010 \pm 0.0156$ |
| Unique ($\uparrow$) | $100.00 \pm 0.00$ | $100.00 \pm 0.00$ | $99.80 \pm 0.10$ |
| Novel ($\uparrow$) | $100.00 \pm 0.00$ | $100.00 \pm 0.00$ | $100.00 \pm 0.10$ |

### F.3    Qualitative Model Samples

In Figure 4, we present uncurated samples from the different models described in Sec. 5.

## G    Baselines

### G.1    GRAN Hyperparameters

For our experiments on the expanded lobster dataset, we use the hyperparameters provided by Liao et al. (2019) for their own (smaller) lobster dataset. For experiments on the expanded planar graph dataset, we utilize the same hyper-parameter setting but reduce the batchsize to 16. For experiments on the SBM dataset, we further reduce the batchsize to 8 and use 2 gradient accumulation steps. For the experiments on the protein dataset, we utilize the pretrained model provided at `http://www.cs.toronto.edu/~rjliao/model/gran_DD.pth`. We perform inference with a batch size of 20.

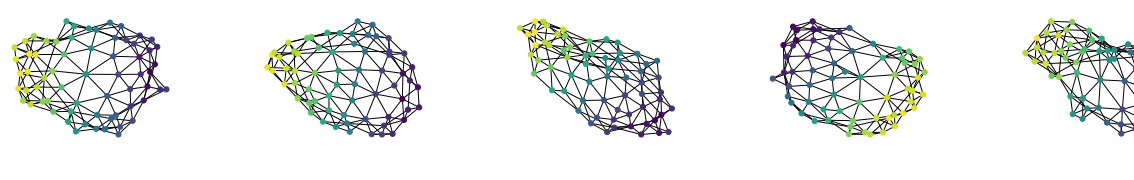

(a) Uncurated samples from ANFM model trained on expanded planar graph dataset.

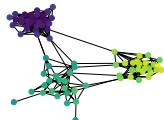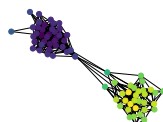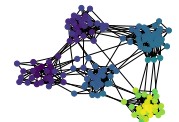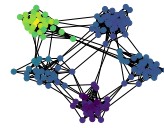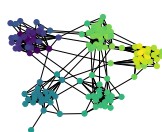

(b) Uncurated samples from ANFM model trained on expanded SBM dataset.

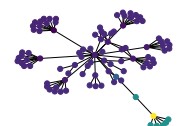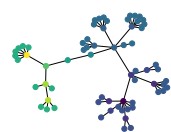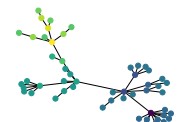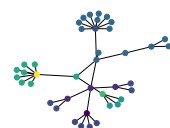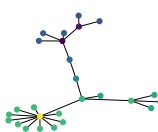

(c) Uncurated samples from ANFM model trained on expanded lobster dataset.

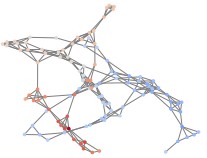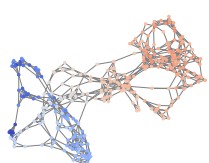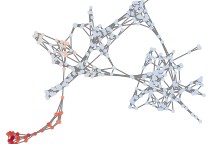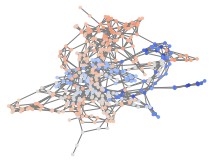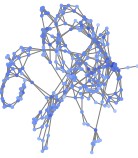

(d) Uncurated samples from ANFM model trained on protein dataset.

Figure 4: Uncurated samples from ANFM (line Fiedler variant).

Table 14: Uniqueness and novelty metrics.

|  | Method | Expaned Planar | Expanded SBM | Expanded Lobster |
|---|---|---|---|---|
| Unique (↑) | GRAN | 100.00 | 100.00 | 99.90 |
|  | DiGress | 100.00 | 100.00 | 99.22 |
|  | ESGG | 100.00 | 100.00 | 99.61 |
|  | ANFM (Fdl.) | 100.00 | 100.00 | 99.80 |
|  | ANFM (DFS) | 100.00 | 100.00 | 99.71 |
| Novel (↑) | GRAN | 100.00 | 100.00 | 97.56 |
|  | DiGress | 100.00 | 100.00 | 96.78 |
|  | ESGG | 100.00 | 100.00 | 98.24 |
|  | ANFM (Fdl.) | 100.00 | 100.00 | 100.00 |
|  | ANFM (DFS) | 100.00 | 100.00 | 99.71 |

### G.2 DiGress Hyperparameters

For our experiments on the expanded planar graph and SBM datasets, we use the hyperparameters provided by Vignac et al. (2023) for the corresponding SPECTRE datasets. On the lobster dataset, we use the same hyperparameters as for the expanded SBM dataset (8 layers and batch size 12). On the protein dataset, we use similar hyperparameters as for the expanded SBM dataset but reduce the batch size to 4 due to GPU memory constraints. We use the same inference approach as Vignac et al. (2023), performing generation with a batch size that is twice as large as the batch size used for training. In all cases, we follow Vignac et al. (2023) in using 1000 diffusion steps.

### G.3 ESGG Hyperparameters

For our experiments on the expanded planar graph and SBM datasets, we use the hyperparameters provided by Bergmeister et al. (2024) for the corresponding SPECTRE datasets. For the expanded lobster dataset, we use the hyperparameters used by Bergmeister et al. (2024) for their tree dataset. We use the test batch sizes provided by Bergmeister et al. (2024) in their hyperparameter configurations.

### G.4 GRAN Model Selection

**Expanded Planar.** In Table 15, we present validation results of the GRAN model trained on the expanded planar graph dataset. We observe no clear development in model performance past 500 steps. We select the checkpoint at 1000 steps.

Table 15: Validation results for GRAN model trained on expanded planar graph dataset. Evaluated on 260 model samples.

| # Steps | Valid (↑) | Node Count (↓) | Degree (↓) | Clustering (↓) | Orbit (↓) | Spectral (↓) |
|---|---|---|---|---|---|---|
| 500 | 0.00 | 0.0065 | 0.0087 | 0.1749 | 0.0693 | 0.0096 |
| 1000 | 0.00 | 0.0007 | 0.0070 | 0.1696 | 0.1100 | 0.0086 |
| 1500 | 0.77 | 0.0100 | 0.0066 | 0.1730 | 0.0743 | 0.0078 |
| 2000 | 0.00 | 0.0021 | 0.0056 | 0.1658 | 0.0816 | 0.0094 |
| 2500 | 0.77 | 0.0033 | 0.0064 | 0.1768 | 0.1042 | 0.0087 |

**Expanded SBM.** In Table 16, we present validation results of the GRAN model trained on the expanded SBM dataset. We find that, overall, the checkpoint at 200 steps appears to perform best and select it.

Table 16: Validation results for GRAN model trained on expanded SBM dataset. Evaluated on 260 model samples.

| # Steps | Valid (↑) | Node Count (↓) | Degree (↓) | Clustering (↓) | Orbit (↓) | Spectral (↓) |
|---------|-----------|----------------|------------|----------------|-----------|--------------|
| 100 | 22.31 | 1.9992 | 0.0243 | 0.0119 | 0.0400 | 0.0037 |
| 200 | 24.23 | 1.9998 | 0.0194 | 0.0114 | 0.0290 | 0.0026 |
| 400 | 20.38 | 1.9999 | 0.0278 | 0.0130 | 0.0448 | 0.0039 |
| 600 | 20.38 | 1.9999 | 0.0225 | 0.0120 | 0.0318 | 0.0030 |

**Expanded Lobster.** In Table 17, we present validation results of the GRAN model trained on the expanded lobster dataset. We observe no improvement in validity past 2500 steps and select this checkpoint.

Table 17: Validation results for GRAN model trained on expanded lobster graph dataset. Evaluated on 260 model samples.

| # Steps | Valid (↑) | Node Count (↓) | Degree (↓) | Clustering (↓) | Orbit (↓) | Spectral (↓) |
|---------|-----------|----------------|------------|----------------|-----------|--------------|
| 500 | 2.34 | 2.0000 | 0.0257 | 0.4753 | 0.2507 | 0.0509 |
| 1500 | 38.67 | 2.0000 | 0.0092 | 0.0112 | 0.1624 | 0.0329 |
| 2500 | 42.58 | 2.0000 | 0.0083 | 0.0059 | 0.1749 | 0.0361 |
| 3500 | 42.97 | 2.0000 | 0.0101 | 0.0049 | 0.1970 | 0.0406 |

### G.5 ESGG Model Selection

While ESGG maintains exponential moving averages of model weights during training, we choose to only evaluate non-smoothed model weights (i.e. the EMA weights with decay parameter $\gamma = 1$), as validation is compute-intensive.

**SBM Dataset.** In our experiments, we obtain worse performance on the expanded SBM dataset than was reported on the smaller SPECTRE SBM dataset in (Bergmeister et al., 2024). In Figure 5, we show the development of validity throughout training, which lasted over 4.5 days on an H100 GPU. Throughout training, we fail to match the validity reported in (Bergmeister et al., 2024). Although the validity estimate is quite noisy, it appears to plateau. We select a model checkpoint at 4.8M steps.

**Protein Dataset.** Model selection on the protein graph dataset is challenging, as the MMD metrics computed during validation are noisy, and generating model samples is time-consuming. We take a structured approach and evaluate model checkpoints at 1-4M training steps using the same validation approach as Bergmeister et al. (2024). Namely, for each graph in the validation set, we generate a corresponding model sample with the same number of nodes. We present the resulting MMD metrics in Table 18. Based

Table 18: Validation results of ESGG trained on protein dataset.

| # Steps | Degree (↓) | Clustering (↓) | Orbit (↓) | Spectral (↓) | Wavelet (↓) | Ratio (↓) |
|---------|------------|----------------|-----------|--------------|-------------|-----------|
| 1M | 0.0242 | 0.1074 | 0.1091 | 0.0095 | 0.0267 | 63.8122 |
| 2M | 0.0028 | 0.0254 | 0.0520 | 0.0009 | 0.0023 | 12.2426 |
| 3M | 0.0066 | 0.0632 | 0.0640 | 0.0030 | 0.0090 | 23.6206 |
| 4M | 0.0293 | 0.1016 | 0.2474 | 0.0079 | 0.0224 | 84.9982 |

on these results, we select the model checkpoint at 2M steps.

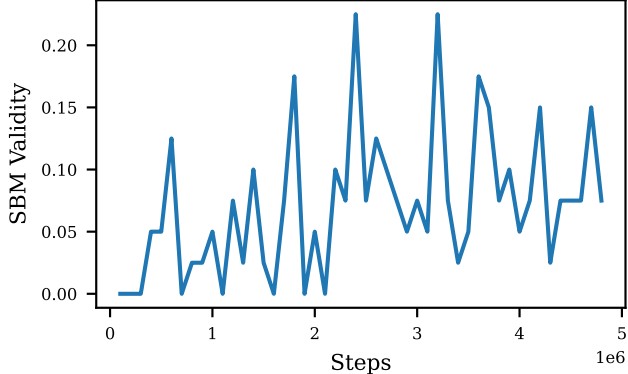

Figure 5: SBM validity during training of ESGG on expanded SBM dataset. Validity is computed using 1000 refinement steps in validation but 100 refinement steps during testing to remain consistent with other baselines.

## H    Additional Ablations

**Noise Augmentation.**    We ablate noise augmentation for the DFS variant of ANFM, supplementing the results previously reported in Table 5. We run 100k training steps of stage I with DFS filtrations on the expanded planar graph dataset and compare performance with and without noise augmentation. Consistent with our previous observations, we find in Table 19 that noise augmentation substantially boosts performance.

Table 19: Performance of DFS variant of AFNM after 100k steps of stage I training on expanded planar graph dataset with and without noise augmentation.

|  | Stage I w/ Noise | Stage I w/o Noise |
|---|---|---|
| VUN ($\uparrow$) | **2.25** | 0.29 |
| Degree ($\downarrow$) | **0.0050** | 0.0381 |
| Clustering ($\downarrow$) | **0.2275** | 0.3136 |
| Spectral ($\downarrow$) | **0.0049** | 0.0169 |
| Orbit ($\downarrow$) | **0.0504** | 0.1045 |

**GAN Tuning.**    We supplement the results on the effectiveness of adversarial finetuning we presented in Table 5. In Table 20, we present ablation results for the DFS variant on the expanded planar graph dataset. In Tables 21 and 22, we compare models after training stage I and II on the expanded SBM and lobster datasets from Sec. 5.2. Again, we observe that adversarial fine-tuning substantially improves performance in terms of validity and MMD metrics.

Table 20: Performance of DFS variant of AFNM after stage I (200k steps) and stage II on expanded planar graph dataset. For results on Fiedler variant, see Table 5.

|  | Stage II | Stage I |
|---|---|---|
| VUN ($\uparrow$) | **45.61** | 2.05 |
| Degree ($\downarrow$) | **0.0003** | 0.0057 |
| Clustering ($\downarrow$) | **0.0296** | 0.2297 |
| Spectral ($\downarrow$) | **0.0017** | 0.0058 |
| Orbit ($\downarrow$) | **0.0004** | 0.0318 |

Table 21: Performance of ANFM models after stage I (200k steps) and stage II on expanded SBM dataset. Showing median ± maximum deviation across three runs for Fiedler variant and a single run for DFS variant. All models attain perfect uniqueness and novelty scores.

| | Fiedler | | DFS | |
| | Stage II | Stage I | Stage II | Stage I |
|---|---|---|---|---|
| VUN (↑) | **75.98** ± 3.71 | 39.65 ± 4.69 | **78.03** | 22.85 |
| Degree (↓) | **0.0014** ± 0.0062 | 0.0023 ± 0.0063 | 0.0008 | **0.0004** |
| Clustering (↓) | **0.0051** ± 0.0009 | 0.0082 ± 0.0012 | **0.0049** | 0.0117 |
| Spectral (↓) | **0.0011** ± 0.0006 | 0.0032 ± 0.0006 | **0.0008** | 0.0020 |
| Orbit (↓) | **0.0180** ± 0.0171 | 0.0210 ± 0.0135 | **0.0049** | 0.0390 |

Table 22: Performance of models after stage I (100k steps) and stage II on expanded lobster dataset. Showing median ± maximum deviation across three runs.

| | Fiedler | | DFS | |
| | Stage II | Stage I | Stage II | Stage I |
|---|---|---|---|---|
| VUN (↑) | **79.10** ± 7.13 | 31.25 ± 4.69 | **87.60** | 47.46 |
| Degree (↓) | 0.0004 ± 0.0013 | **0.0004** ± 0.0010 | $7.82 \times 10^{-5}$ | 0.0209 |
| Clustering (↓) | $\mathbf{7.89 \times 10^{-5}}$ ± $5.32 \times 10^{-5}$ | 0.0136 ± 0.0054 | $\mathbf{1.64 \times 10^{-6}}$ | 0.0043 |
| Spectral (↓) | **0.0016** ± 0.0028 | 0.0030 ± 0.0012 | **0.0010** | 0.0119 |
| Orbit (↓) | **0.0010** ± 0.0156 | 0.0073 ± 0.0026 | **0.0007** | 0.0206 |
| Unique (↑) | **99.80** ± 0.10 | 99.51 ± 0.39 | 99.71 | **100.00** |
| Novel (↑) | **100.00** ± 0.10 | 99.90 ± 0.39 | 99.71 | **99.90** |

**Filtration Function.** In Table 23, we study alternative filtration functions. We compare the line fiedler function to centrality-based filtration functions. Following Anthonisse (1971); Brandes (2008), we let $\sigma(i, j)$ denote the number of shortest paths between two nodes $i, j \in V$, and $\sigma(i, j \,|\, e)$ denote the number of these paths passing through an edge $e \in E$. Then, we define the betweenness centrality function as:

$$f_{\text{between}}(e) := \sum_{i,j \in V} \frac{\sigma(i, j | e)}{\sigma(i, j)}, \qquad \forall\, e \in E. \tag{33}$$

Based on this, we define the remoteness centrality as $f_{\text{remote}}(e) = -f_{\text{between}}(e)$. We use the same filtration scheduling approach as for the line Fiedler function. We observe that the line fiedler function appears to out-perform the two alternatives in our setting.

Table 23: Performance after training stage I with different filtration functions for 100k steps on expanded planar graph dataset. Showing median of three runs for spectral variant and one run each for betweenness and remoteness variants.

| | VUN (↑) | Degree (↓) | Clustering (↓) | Spectral (↓) | Orbit (↓) |
|---|---|---|---|---|---|
| Line Fiedler | **20.21** | 0.0058 | **0.1768** | **0.0048** | **0.0129** |
| DFS | 2.25 | **0.0050** | 0.2275 | 0.0049 | 0.0504 |
| Betweenness | 0.20 | 0.0069 | 0.2724 | 0.0124 | 0.0804 |
| Remoteness | 3.52 | 0.0136 | 0.2720 | 0.0085 | 0.0234 |

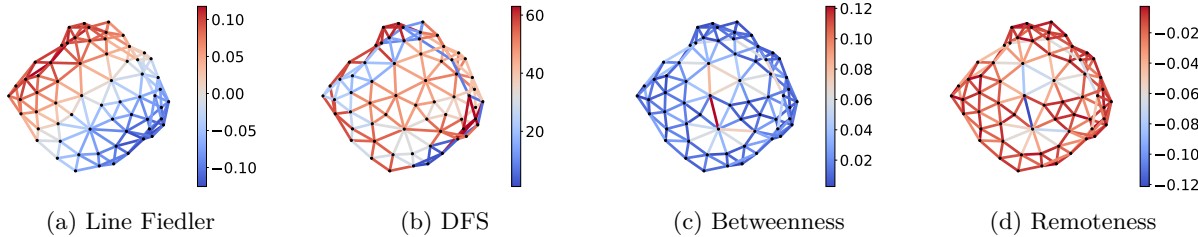

|            (a) Line Fiedler           |           (b) DFS            |         (c) Betweenness         |          (d) Remoteness          |

Figure 6: Visualization of different filtration functions on a planar graph

**Scheduling.** We recall that the filtration schedule of the line Fiedler variant depends on a mononotonically increasing function $\gamma : [0, 1] \to [0, 1]$. Here, we study the performance of the three choices for $\gamma$:

$$
\begin{aligned}
\text{Linear}: \quad & \gamma(t) := t \\
\text{Convex}: \quad & \gamma(t) := 1 - \cos\left(\frac{\pi t}{2}\right) \\
\text{Concave}: \quad & \gamma(t) := \sin\left(\frac{\pi t}{2}\right)
\end{aligned}
\tag{34}
$$

We present results on the planar graph dataset in Table 24. We find that no single variant performs

Table 24: Performance after training stage I with different filtration schedules for 100k steps on expanded planar graph dataset with line Fiedler variant. All models attain perfect uniqueness and novelty scores. Showing median of three runs for linear variant and one run each for convex and concave variants.

|          | VUN ($\uparrow$) | Degree ($\downarrow$) | Clustering ($\downarrow$) | Spectral ($\downarrow$) | Orbit ($\downarrow$) |
|----------|--------|--------|------------|----------|--------|
| Linear   | 20.21  | 0.0058 | 0.1768     | 0.0048   | 0.0129 |
| Convex   | 5.66   | **0.0043** | 0.2239 | **0.0040** | **0.0062** |
| Concave  | **31.05** | 0.0045 | **0.1590** | 0.0059 | 0.0153 |

consistently best across all evaluation metrics. However, the concave variant attains the highest validity score.

**Node Individualization.** In Table 25, we study different node individualization techniques for the line Fiedler variant of ANFM. We refer to the ordering scheme we describe in Appendix E.1 as the *derived ordering*, as it is based on the values of the line Fiedler filtration function. Additionally, we study *random orderings* and node orderings according to a *depth first search (DFS)*. Moreover, we compare to node individualizations that do not consist of positional embeddings w.r.t. a node orderings but instead i.i.d. *gaussian noise* that is re-applied in each time-step. Finally, we also consider a variant in which no individualization is applied, i.e., the embedding matrix $W^{\text{node}}$ is fixed to be all-*zeros*. We find that individualizing nodes with learned embeddings based on some ordering (either random, derived from the line Fiedler filtration function, or a DFS search) appears to be beneficial. On the planar graph dataset, there is no clear benefit of the derived ordering over random orderings. However, we observe a clear advantage on the SBM dataset, as can be seen in Table 26.

Table 25: Performance after training stage I of the line Fiedler variant with different node individualization techniques for 100k steps on expanded planar graph dataset. All models attain perfect uniqueness and novelty scores. Showing median of three runs for derived ordering and one run each for all other variants.

|  | VUN ($\uparrow$) | Degree ($\downarrow$) | Clustering ($\downarrow$) | Spectral ($\downarrow$) | Orbit ($\downarrow$) |
|---|---|---|---|---|---|
| Derived Ordering | 20.21 | 0.0058 | **0.1768** | 0.0048 | 0.0129 |
| DFS Ordering | **28.91** | **0.0055** | 0.1803 | 0.0024 | **0.0053** |
| Random Ordering | 18.36 | 0.0085 | 0.2332 | **0.0023** | 0.0091 |
| Gaussian Noise | 12.99 | 0.0086 | 0.2356 | 0.0031 | 0.0112 |
| Zeros | 13.48 | 0.0057 | 0.2195 | 0.0023 | 0.0091 |

Table 26: Performance after training stage I of the line Fiedler variant with derived and random node ordering after 100k steps on expanded SBM datasets. Showing median $\pm$ maximum deviation across three runs for derived ordering and one run for random ordering.

|  | Derived Ordering | Random Ordering |
|---|---|---|
| VUN ($\uparrow$) | **26.95** $\pm$ 2.64 | 2.44 |
| Degree ($\downarrow$) | **0.0222** $\pm$ 0.0127 | 0.0396 |
| Clustering ($\downarrow$) | **0.0106** $\pm$ 0.0012 | 0.0122 |
| Spectral ($\downarrow$) | **0.0061** $\pm$ 0.0014 | 0.0144 |
| Orbit ($\downarrow$) | **0.0548** $\pm$ 0.0244 | 0.0596 |
| Unique ($\uparrow$) | **1.0000** $\pm$ 0.0000 | 0.9951 |
| Novel ($\uparrow$) | **1.0000** $\pm$ 0.0000 | **1.0000** |

## I Bias and Variance of Estimators

Previous works (Martinkus et al., 2022; Vignac et al., 2023; Bergmeister et al., 2024) evaluate their graph generative models on as few as 40 samples. In this section, we investigate how this practice impacts the variance and bias of the estimators used in model evaluation and argue that a higher number of test samples should be chosen.

### I.1 Variance of Validity Estimation

On synthetic datasets such as those introduced in (Martinkus et al., 2022), one may verify whether model samples are "valid", i.e., whether they satisfy a property that is fulfilled by (almost) all samples of the true data distribution. By taking the ratio of valid graphs out of $n$ model samples, previous works have estimated the probability of obtaining valid graphs from the generator.

**Definition I.1.** Let the random variable $G$ denote a sample from a graph generative model and let valid : $\mathcal{G} \to \{0,1\}$ a measurable binary function that determines whether a sample is valid. Then the models true validity ratio is defined as:

$$\mathbb{P}[\text{valid}(G) = 1] \tag{35}$$

For i.i.d. samples $G_1, \ldots, G_n$, we introduce the following estimator:

$$V := \frac{\sum_{i=1}^{n} \text{valid}(G_i)}{n} \tag{36}$$

Given the simplicity of the validity metric, we can very easily derive the uncertainty of the estimator used for evaluation. We make this concrete in Proposition I.2.

**Proposition I.2.** *For a generative model with a true validity ratio of $p \in [0,1]$, the validity estimator on $n$ samples is unbiased and has standard deviation $\sqrt{p(1-p)}/\sqrt{n}$.*

*Proof.* Assuming that the random variables $G_1, \ldots, G_n$ are i.i.d. samples from the generative model, then the random variables $\text{valid}(G_1), \ldots, \text{valid}(G_n)$ are i.i.d. according to Bernoulli$(p)$. The validity estimator is given as:

$$V = \frac{\sum_{i=1}^n \text{valid}(G_i)}{n} \tag{37}$$

By the linearity of expectation, we have

$$\mathbb{E}[V] = \frac{\sum_{i=1}^n \mathbb{E}[\text{valid}(G_i)]}{n} = \frac{np}{p} = p \tag{38}$$

which shows that the estimator is unbiased. The variance is given by:

$$\text{Var}[V] = \frac{\text{Var}\left[\sum_{i=1}^n \text{valid}(G_i)\right]}{n^2} = \frac{\sum_{i=1}^n \text{Var}[\text{valid}(G_i)]}{n^2}$$
$$= \frac{p(1-p)}{n} \tag{39}$$

where we used the independence assumption in the first line. Taking the square root, we obtain the standard deviation from the proposition. $\square$

From Proposition I.2, we note that the standard deviation of the validity estimate can be as high as $1/(2\sqrt{n})$, which is achieved at $p = 0.5$. For $n = 40$, we find that the standard deviation can therefore be as high as 7.9 percentage points.

We illustrate the magnitude of this effect on a practical example: for DiGress, we obtain a validity ratio on the SBM dataset of approximately $p \approx 56.2\%$ (c.f., Table 2). While this estimated ratio itself is subject to uncertainty, we assume it to be the ground truth for the sake of the following illustration (justified by the large sample size of 1024 graphs used to estimate it). In Table 27, we analytically (using Proposition I.2) compute the mean and standard deviation of the validity estimator at different sample sizes. We note that the mean is constant since the estimator is unbiased. However, the standard deviation remains substantial, even at 128 graph samples.

Table 27: Mean validity estimate $\pm$ standard deviation across evaluations of validity estimator. Computed analytically via Proposition I.2. The estimated validity is unbiased but has a high standard deviation.

| # Model Samples | Validity Estimate |
|---|---|
| 32 | $56.2 \pm 8.8$ |
| 64 | $56.2 \pm 6.2$ |
| 128 | $56.2 \pm 4.4$ |
| 256 | $56.2 \pm 3.1$ |
| 512 | $56.2 \pm 2.2$ |
| 1024 | $56.2 \pm 1.6$ |

## I.2 Bias and Variance of MMD Estimation

**Definition I.3.** Let $(\mathcal{X}, d)$ be a metric space and let $k : \mathcal{X} \times \mathcal{X} \to \mathbb{R}$ be a measurable, symmetric kernel which is bounded but not necessarily positive-definite. Let $X := [x_1, \ldots, x_n]$ be i.i.d. samples from a Borel distribution $p_x$ on $\mathcal{X}$ and $Y := [y_1, \ldots, y_n]$ be i.i.d. samples from a distribution $p_y$. Assume $X$ and $Y$ to be independent. Following (Gretton et al., 2012), define the squared MMD of $p_x$ and $p_y$ as:

$$\text{MMD}^2(p_x, p_y) := \mathbb{E}[k(x_1, x_2)] + \mathbb{E}[k(y_1, y_2)] - 2\mathbb{E}[k(x_1, y_1)] \tag{40}$$

and note that this is well-defined by our assumptions. Finally, introduce the following estimator for the squared MMD:

$$M := \frac{1}{n^2} \sum_{i,j=1}^n k(x_i, x_j) + \frac{1}{m^2} \sum_{i,j=1}^m k(y_i, y_j) - \frac{2}{nm} \sum_{i=1}^n \sum_{j=1}^m k(x_i, y_j) \tag{41}$$

We empirically study bias and variance of the MMD estimates on the planar graph dataset. We generate 8192 samples from one of our trained models and repeatedly compute the MMD between the test set and a random subset of those samples. We vary the size of the random subsets and run 64 evaluations for each size, computing the mean and standard deviation of the MMD metrics across the 64 evaluations. We report the results in Table 28. We observe that, on average, the MMD is severely over-estimated when using fewer

Table 28: Mean MMD $\pm$ standard deviation across 64 evaluation runs of a single model. The test set contains 256 planar graphs, while a varying number of model samples is used, as indicated on the left. The MMD and its variance decrease substantially with larger numbers of model samples.

| # Model Samples | Degree ($\downarrow$) | Clustering ($\downarrow$) | Spectral ($\downarrow$) |
|---|---|---|---|
| 32 | $8.59 \times 10^{-4} \pm {\scriptstyle 5.59 \times 10^{-4}}$ | $4.21 \times 10^{-2} \pm {\scriptstyle 1.44 \times 10^{-2}}$ | $4.73 \times 10^{-3} \pm {\scriptstyle 9.14 \times 10^{-4}}$ |
| 64 | $5.58 \times 10^{-4} \pm {\scriptstyle 2.90 \times 10^{-4}}$ | $2.68 \times 10^{-2} \pm {\scriptstyle 7.58 \times 10^{-3}}$ | $2.59 \times 10^{-3} \pm {\scriptstyle 4.78 \times 10^{-4}}$ |
| 128 | $4.40 \times 10^{-4} \pm {\scriptstyle 1.79 \times 10^{-4}}$ | $2.17 \times 10^{-2} \pm {\scriptstyle 4.33 \times 10^{-3}}$ | $1.61 \times 10^{-3} \pm {\scriptstyle 3.16 \times 10^{-4}}$ |
| 256 | $4.39 \times 10^{-4} \pm {\scriptstyle 1.45 \times 10^{-4}}$ | $2.02 \times 10^{-2} \pm {\scriptstyle 3.89 \times 10^{-3}}$ | $1.14 \times 10^{-3} \pm {\scriptstyle 1.86 \times 10^{-4}}$ |
| 512 | $4.32 \times 10^{-4} \pm {\scriptstyle 8.48 \times 10^{-5}}$ | $1.81 \times 10^{-2} \pm {\scriptstyle 2.56 \times 10^{-3}}$ | $1.18 \times 10^{-3} \pm {\scriptstyle 2.04 \times 10^{-4}}$ |
| 1024 | $4.26 \times 10^{-4} \pm {\scriptstyle 5.99 \times 10^{-5}}$ | $1.72 \times 10^{-2} \pm {\scriptstyle 1.66 \times 10^{-3}}$ | $1.18 \times 10^{-3} \pm {\scriptstyle 2.00 \times 10^{-4}}$ |

than 256 model samples. At the same time, the variance between evaluation runs is large when few samples are used, making the results unreliable.

## J  Adversarial Finetuning Details

We provide pseudocode for the adversarial fine-tuning stage in Algorithm 1. We note that we do not make all procedures explicit and that many hyper-parameters must be chosen (including the number of steps and epochs in TRAINGENERATORANDVALUEMODEL).

**Generator.**  The generator operates in inference mode, meaning that all dropout layers are disabled and batch normalization modules utilize the (now frozen) moving averages from training stage I. Hence, the behavior of the generative model becomes reproducible. It acts as a stochastic policy in a higher-order MDP, where the graphs $G_0, \ldots, G_T$ are the states. It receives a terminal reward for the plausibility of the final sample $G_T$.

**Discriminator.**  The discriminator is implemented as a GraphGPS (Rampásek et al., 2022) model which performs binary classification on graph samples $G_T$, distinguishing real samples from generated samples. It is trained via binary cross-entropy on batches consisting in equal proportions of generated graphs and graphs from the dataset $\mathcal{D}$. For a given graph $G_T$, the discriminator produces a probability of "realness" by applying the sigmoid function to its logit. Following SeqGAN (Yu et al., 2017), the log-sigmoid of the logit then acts as a terminal reward for the generative model. We emphasize that only the final graph $G_T$ is presented to the discriminator.

**Value Model.**  The value model uses the same backbone architecture as our generative model and regresses scalars from pooled node representations. It is trained via least squares regression. The value model is used to compute baselined reward-to-go values.

**Training Outline.**  While Algorithm 1 provides a technical description of the training algorithm, we also provide a rougher outline here. At the start of training stage II, the generator is initialized with the weights learned in training stage I, while the discriminator and value model are initialized randomly. Before entering the main training loop, we pre-train the discriminator and value model to match the generator. Namely, we first pre-train the discriminator to classify graphs as either "real" or "generated". The log-likelihood of "realness" acts as a terminal reward of the generative model. The discriminator is then pre-trained to regress the reward-to-go. After pre-training is finished, we proceed to the training loop, which consists of alternating training of (i) the generator and value model and (ii) the discriminator. As described above, the generator

is trained via PPO to maximize the terminal reward provided by the discriminator. The value model is used to baseline the reward and is continuously trained to regress the reward-to-go. The discriminator, on the other hand, continues to be trained on generated and real graph samples via binary cross-entropy.

---

**Algorithm 1** Adversarial Finetuning

---

**procedure** TRAINGENERATORANDVALUEMODEL($p_\theta$, $d_\varphi$, $v_\vartheta$)
    **for** $i = 1 \ldots, N_{\text{steps}}$ **do**
        $\mathcal{S} \leftarrow [\,]$                                                   ▷ List of sampled filtrations
        $r \leftarrow 0 \in \mathbb{R}^{N_{\text{samples}}}$                                       ▷ Terminal rewards
        **for** $j = 1 \ldots, N_{\text{samples}}$ **do**
            $G_0^{(j)}, \ldots, G_T^{(j)} \leftarrow$ SAMPLEFILTRATION($p_\theta$)
            $\mathcal{S}.\text{append}\left(\left(G_0^{(j)}, \ldots, G_T^{(j)}\right)\right)$
            $r_j \leftarrow \text{logsigmoid}(d_\varphi(G_T^{(j)}))$
            $r_j \leftarrow \max(r_j, R_{\text{lower}})$                             ▷ Reward clamping
        **end for**
        $r \leftarrow$ WHITEN($r$)                      ▷ Whiten rewards using EMA of mean and std
        $g_{j,t} \leftarrow 0 \qquad \forall j = 1, \ldots, N_{\text{samples}} \,\forall t = 0, \ldots, T-1$       ▷ Rewards-to-go
        **for** $j = 1 \ldots, N_{\text{samples}}$ **do**
            **for** $t = 0, \ldots, T-1$ **do**
                $g_{j,t} \leftarrow r_j - v_\vartheta(G_0^{(j)}, \ldots, G_t^{(j)})$                 ▷ Compute baselined RTG
            **end for**
        **end for**
        TRAINVALUEMODEL($v_\vartheta, \mathcal{S}, r$)
        **for** $k = 1 \ldots, N_{\text{epoch}}$ **do**
            $l_{j,t}^{(k)} \leftarrow -\log p_\theta(G_t^{(j)}|G_{t-1}^{(j)}, \ldots, G_0^{(j)}) \qquad \forall j = 1, \ldots, N_{\text{samples}} \quad \forall t = 1, \ldots, T$
            $u_{j,t} \leftarrow \exp(\text{sg}[l_{j,t}^{(1)}] - l_{j,t}^{(k)}) \qquad \forall j, t$
            $\mathcal{L}_{j,t}^{(1)} \leftarrow -u_{j,t} \cdot g_{j,t-1} \qquad \forall j, t$
            $\mathcal{L}_{j,t}^{(2)} \leftarrow -\text{clamp}(u_{j,t}, 1-\epsilon, 1+\epsilon) \cdot g_{j,t-1} \qquad \forall j, t$
            $\mathcal{L} \leftarrow \sum_{j,t} \max(\mathcal{L}_{j,t}^{(1)}, \mathcal{L}_{j,t}^{(2)})$
            $\theta \leftarrow \theta - \delta \nabla_\theta \mathcal{L}$                        ▷ Backpropagate and update parameters
        **end for**
    **end for**
**end procedure**

**procedure** GANTUNING($p_\theta$, $\mathcal{D}$)            ▷ Takes generator from training stage I and graph dataset
    $d_\varphi \leftarrow$ new GNN                                            ▷ Initialize discriminator
    TRAINDISCRIMINATOR($p_\theta, d_\varphi, \mathcal{D}$)                         ▷ Pre-train discriminator
    $v_\vartheta \leftarrow$ new mixer model
    $\mathcal{S} \leftarrow$ GENERATEFILTRATIONS($p_\theta$)
    $r \leftarrow$ GRADESAMPLES($\mathcal{S}, d_\varphi$)
    TRAINVALUEMODEL($v_\vartheta, \mathcal{S}, r$)                              ▷ Pre-train value model
    **while** not converged **do**
        TRAINGENERATORANDVALUEMODEL($p_\theta, d_\varphi, v_\vartheta$)
        TRAINDISCRIMINATOR($p_\theta, d_\varphi, \mathcal{D}$)
    **end while**
**end procedure**

---

## K   Training Costs

In Table 29, we compare the computational costs of training the presented ANFM models (line Fiedler variant) to the costs of training the presented DiGress models. We note that the preparation of filtrations for the first training stage is CPU-bound and highly parallelizable. In comparison to the cost of training, it is negligible.

Table 29: Training costs of ANFM and DiGress across datasets and training stages.

| Dataset | ANFM | | | DiGress |
|---|---|---|---|---|
| | Stage 1 | Stage 2 | Total H100 Hours | Total H100 Hours |
| Expanded Planar | 2 H100s × 20h | 1 H100 × 20h | 60h | 95h |
| Expanded SBM | 2 H100s × 20h | 1 H100 × 81h | 121h | 90h |
| Expanded Lobster | 2 H100s × 12h | 1 H100 × 58h | 82h | 48h |

We find that training of DiGress is often cheaper than training of ANFM. Hence, ANFM trades efficiency during inference for increased computational costs during training.

## L   MMD vs Filtration Granularity

In this section, we supplement the results from Figure 2, illustrating how the MMD metrics evolve as the number of generation steps is varied in ANFM and DiGress. Figure 7 shows that, in terms of MMD, ANFM generates higher-quality graphs than DiGress at the considered number of generation steps. This is consistent with our findings from Figure 2. We recall that the default DiGress variant uses 1000 generation steps.

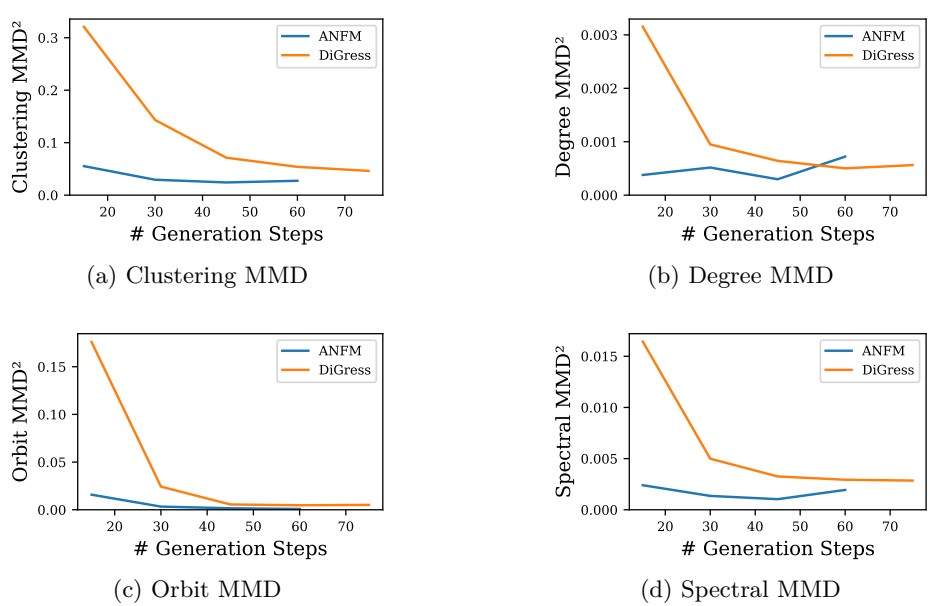

(a) Clustering MMD

(b) Degree MMD

(c) Orbit MMD

(d) Spectral MMD

Figure 7: Performance of ANFM and DiGress in terms of MMD on the expanded planar dataset as the number of generation steps varies. The original (default) DiGress variant uses 1000 generation steps.

## M   Detailed Definition of the Line Fiedler Filtration Function

In this section, we provide a more thorough definition of the line Fiedler filtration function we have introduced in Sec. 4.1. Let $G = (V, E)$ be an undirected connected graph on the nodes $V$ with edges $E \subset V \times V$. We

define its line graph as $L(G) := (E, \boldsymbol{E}')$ where $\boldsymbol{E}'$ denotes the pairs of edges that share a vertex in $G$:

$$\boldsymbol{E}' := \{(e, e') \ : \ e, e' \in E, \ |e \cap e'| = 1\} \tag{42}$$

It is easy to verify that $L(G)$ is connected. Let $A \in \mathbb{R}^{E \times E}$ be its adjacency matrix and define the associated symmetrically normalized Laplacian matrix as:

$$L := I - D^{-\frac{1}{2}} A D^{-\frac{1}{2}}, \tag{43}$$

where $D \in \mathbb{R}^{E \times E}$ is the diagonal matrix with $D_{i,i} = \sum_{j \in E} A_{i,j}$. We may now find the eigenvectors associated to the second smallest eigenvalue. These are the Fiedler vectors of $L(G)$. One such vector $v \in \mathbb{R}^E$ with unit length provides the line Fiedler filtration function. We note that, similar to the DFS filtration function, the line Fiedler filtration function is not unique for a given graph. Indeed, the Fiedler vector of the line graph is at most unique up to a sign.

