# OpenReview forum: "Fast Graph Generation via Autoregressive Noisy Filtration Modeling"
_TMLR — Accepted by TMLR_

### Review · Reviewer_3vAw · 2025-10-06

**Summary Of Contributions:**

### Sumary
This work introduces the Autoregressive Noisy Filtration Model (ANFM), an autoregressive framework for graph generation.

The core innovation is its use of graph filtrations, a concept from topological data analysis, to transform a target graph into a concise sequence of increasingly complex subgraphs. The procedure generates a short sequence of increasingly dense and detailed intermediate graphs, which interpolate the target graph and the fully disconnected graph. The authors compellingly argue that this filtration-based approach generalizes prior autoregressive methods by naturally enabling the construction of non-induced subgraphs.

The proposed ANFM is based on a two-stage training procedure designed to address challenges in generative modeling:

- Stage I employs noise augmentation, which allows for both the addition and deletion of nodes and edges. This creates non-monotonic graph sequences.
- Stage II further refines the model using a reinforcement learning (RL) based fine-tuning approach to improve sample quality.

Presented results suggest that ANFM achieves a favorable trade-off between sample quality and inference speed, as ANFM can match the performance of state-of-the-art graph diffusion models while offering (in practice) faster generation times.

The **main strengths of the work** consist of:

- The idea of employing graph filtrations for autoregressive generation. The theoretical insight that this framework generalizes prior methods to allow for non-induced subgraphs is a contribution.

- The authors identify exposure bias as a primary hurdle for autoregressive graph generation models and propose a noise augmentation framework in Stage I and an adversarial RL fine-tuning approach in Stage II as solutions to mitigate this issue.
    - The ablation study presented in Section 5.4 provides empirical evidence for the importance of both components.
    - This validates the effectiveness of their proposed techniques.

Despite these strengths, there are **a few weaknesses** to be clarified or strengthened:

- My main concern is a potential disconnect between the proposed novelty (filtration) and the demonstrated sources of performance.
    - While filtration is an elegant framing, the empirical results, particularly the ablation studies, suggest that the noise-augmentation and RL fine-tuning components are the primary drivers of the model's success. It remains unclear how much of the performance gain is directly attributable to the filtration-based sequence generation itself.

- The analysis of computational complexity feels incomplete. The paper focuses on asymptotic inference complexity, arguing that a small number of steps (T) makes the method efficient.
    - However, the training cost, which may be substantial ---due to filtration sequence creation, the main training loop, and the subsequent RL fine-tuning--- is not discussed.
    - A more holistic view of the computational trade-offs seems warranted.

**Audience:**

Yes

**Audience Explanation:**

This work presents both a practical solution to the problem of graph generation and a new theoretical framework based on filtration.

What makes the work interesting is the core generative mechanism: a novel framework based on graph filtrations, which the authors show is a formal generalization of prior autoregressive techniques.

The authors effectively tackle the challenge of exposure bias in autoregressive models via a two-stage method of noise augmentation and RL fine-tuning, backed by ablation studies.

**Claims And Evidence:**

No

**Claims Explanation:**

See the weakness described above, mainly pertaining to disentangling the sources of performance (i.e, how much of the performance gain is directly attributable to the filtration-based sequence generation itself) and a more detailed analysis of the computational trade-offs between training and inference.

**Requested Changes:**

1. The most critical point is to better isolate and articulate the precise contribution of the filtration framework to the final performance. Currently, its benefits seem overshadowed by the noise augmentation and RL components. Could the authors elaborate on this interplay?
    - A compelling way to disentangle these effects would be to conduct an experiment comparing ANFM to a variant that uses a more traditional autoregressive scheme (e.g., node-by-node addition) but still incorporates the same noise-augmentation and RL fine-tuning stages.

2. The discussion on computational cost should be expanded to provide a complete picture.
    1. Please report on the training costs. This should include a discussion and empirical reporting of the time required for creating graph filtration sequences, the main Stage I training, and the Stage II RL fine-tuning.
    2. Regarding inference, can you provide empirical data on how execution time scales with T, rather than just the asymptotic analysis?
        - In the Filtration Granularity analysis (Figure 2), why is T limited to 45?
        - A thorough analysis (with comparison to diffusion-based baselines) showing the trade-off between sample quality (all metrics) and inference time as T increases would be very valuable.
    3. The reporting of "time needed to generate 1024 graphs... normalized to a per-graph cost" is sensitive to implementation details. Please provide more context or justification for why this is a robust measure.

3. To strengthen the submission, please discuss the following:
    1. The authors identify SID (Boget, 2025) as a highly relevant discrete diffusion model, but it is not included as a baseline. Please clarify the reason for its omission. Comparing against this work would further situate ANFM in the current state-of-the-art landscape.
    2. Expand on the importance of the filtration design choices and hyperparameters. For instance:
        - A discussion on the choice of a fixed T for all graphs in a dataset would be beneficial. Have the authors considered whether an adaptive T (e.g., dependent on graph size or complexity) could be advantageous?
        - The appendix contains a valuable investigation of different filtration strategies. A summary of the key takeaways ---highlighting the benefits and drawbacks of each strategy--- in the main paper would be an excellent addition for the reader.

4. Minor Suggestions for Presentation and Clarity:
    1. Given its centrality to the paper, consider adding a slightly more detailed background explanation of graph filtrations in the main text, to improve accessibility for a broader audience.
    2. Please provide explicit inline definitions for key terms when they are first introduced (e.g., "induced subgraphs," "monotonic graph sequences").
    3. On all result tables, please use bolding to highlight the best-performing method for each metric, and perhaps underlining for the best in each model class (e.g., best autoregressive, best diffusion). This would greatly aid in the rapid interpretation of results.

---

> ### Author Response · Authors · 2025-10-18
> **Response to Reviewer 3vAw**
>
> **Edit: We posted our response on Oct. 18, and just realized that our response is not visible to you. We are really sorry for any inconvenience.**
>
> Thank you for your time and effort in providing valuable feedback! Below, we provide initial responses to your requested changes (RC). We will update our manuscript as soon as we receive all reviews.
>
> ### Isolate and articulate the precise contribution of the filtration framework (RC 1)
>
> We appreciate the opportunity to elaborate on the interplay between our proposed filtration framework and the noise augmentation and RL components.
>
> Compared to *node-addition paradigms*, our filtration framework introduces two new abilities:
>
> 1. It can model non-monotonic sequences of graphs (allowing both edge additions and deletions).
> 2. It can accommodate a more flexible family of graph sequences (e.g., non-induced subgraphs via line Fiedler filtrations).
>
> **Crucially, the first property is necessary for noise augmentation.** Noise augmentation requires exposing the model to both edge addition and edge deletion operations, which necessitates non-monotonic sequence modeling. Traditional node-addition methods, such as GraphRNN and GRAN, fundamentally cannot model non-montonic sequences, making noise augmentation inapplicable without our filtration formulation.
>
> As noted in Section 4.1, node-addition strategies can be emulated by ANFM, e.g., via the DFS filtration strategy. We studied the effect of noise augmentations with this filtration strategy in Appendix H (Table 19), and consistent with our main findings, we confirmed that noise augmentation provides significant benefits in this setting.
>
> Regarding RL fine-tuning: while this technique may be applicable to traditional autoregressive paradigms like GRAN, implementing it would require substantial extensions to these methods, which fall outside the scope of our work.
>
> **Proposed manuscript changes:**
>
> - We will explicitly emphasize that modeling non-monotonic sequences is a necessary prerequisite for noise augmentations.
> - We will add a discussion noting that existing autoregressive paradigms may benefit from RL fine-tuning, though this requires non-trivial modifications.
> - We will strengthen the presentation of how filtration enables both of our proposed techniques.
>
> ### Training cost (RC 2.1)
>
> Below, we provide approximately estimated training times on the expanded synthetic datasets:
>
> | Dataset          | Stage 1        | Stage 2      | Total H100 hours |
> |------------------|----------------|--------------|------------------|
> | Expanded Planar  | 2 H100s x 20h  | 1 H100 x 20h | 60h              |
> | Expanded SBM     | 2 H100s x 20h  | 1 H100 x 81h | 121h             |
> | Expanded Lobster | 2 H100s x 12h  | 1 H100 x 58h | 82h              |
>
> The computational cost of constructing graph filtrations is negligible in comparison: this process is CPU-bound, relatively inexpensive, and easily parallelizable. We can provide detailed timing measurements if desired.
>
> For comparison, we also provide the training times for the DiGress models we present in the paper:
>
> | Dataset          | Digress H100 hours |
> |------------------|--------------------|
> | Expanded Planar  | 95h                |
> | Expanded SBM     | 90h                |
> | Expanded Lobster | 48h                |
>
> Overall, ANFM models are more expensive to train than DiGress models. In most cases, the adversarial finetuning is the computationally most expensive part.
>
> We will incorporate these results in the revised manuscript.
>
> ### Inference cost w.r.t. T (RC 2.2)
>
> Please note that we already included empirical measurements of inference time as a function of T in Figure 2b.
>
> **Regarding the T=45 limit in Figure 2:** This choice was made to balance sample quality and computational cost during our experiments. We will extend our figure to include larger T values.
>
> **Trade-off between sample quality and T:** We will add other MMD metrics as a function of T to the appendix.
>
> ### Measurement of inference cost (RC 2.3)
>
> We completely agree that measured inference times depend on implementational details. We conducted these measurements in good faith, ensuring all models were evaluated on identical hardware under comparable conditions. For DiGress, we additionally explored whether reducing the number of denoising iterations would make it competitive with ANFM (cf. Figure 2).
>
> While implementation factors cannot be entirely eliminated, we believe our measurements provide a reasonable comparison. We will update the manuscript to explicitly discuss this limitation and provide additional context on our experimental setup to improve reproducibility and interpretation.

---

> ### Author Response · Authors · 2025-10-18
> **Response to Reviewer 3vAw (cont.)**
>
> ### Related work: SID (RC 3.1)
>
> We did not include SID (Boget, 2025) as a baseline for several reasons:
>
> - SID was not evaluated on most of our considered datasets, which would require extensive hyperparameter tuning and adaptation to ensure fair comparison. Given the substantial effort required to properly tune and validate SID across our diverse set of benchmarks (including the expanded synthetic datasets, Proteins, and GuacaMol), we prioritized direct comparisons with well-established baselines.
> - The evaluation in the SID paper was relatively limited in scope, making it difficult to conclusively determine whether SID outperforms well-established diffusion models such as DiGress, which already serves as a strong baseline in our work. Since DiGress represents the current state-of-the-art among diffusion-based approaches and we demonstrate competitive performances, we believe our comparisons adequately position ANFM within the landscape of graph generation methods and **our current experiments sufficiently support all our claims in the manuscript**.
> Nevertheless, we agree that SID represents an important contribution to discrete diffusion for graphs. We will extend our discussion of SID to highlight the conceptual differences with our proposed approach.
>
>
> ### filtration design choices and hyperparameters (RC 3.2)
>
> > Have the authors considered whether an adaptive T (e.g., dependent on graph size or complexity) could be advantageous?
>
> We did consider using adaptive sequence lengths (dependent on graph size or complexity), as is common in autoregressive models like GraphRNN. However, we identified several significant disadvantages:
>
> - The model would require an "end of sequence" token, which complicates both architecture and training.
> - Training and inference potentially become more expensive due to ragged sequences requiring padding, while a fixed $T$ allows for better batching efficiency.
> - Determining the appropriate sequence length is non-trivial; using one step per edge would substantially increase computational costs.
>
> We will add this discussion to the manuscript.
>
>
> > The appendix contains a valuable investigation of different filtration strategies. A summary [...] in the main paper would be an excellent addition
>
> Thank you for this excellent suggestion. We will add a concise summary table to the main paper, highlighting the key benefits and drawbacks of each filtration strategy investigated in the appendix, helping readers quickly understand when each approach is most appropriate.
>
> ### Minor presentation improvements (RC 4)
>
> We will implement all your suggestions for improving clarity and accessibility.

---

> ### Comment · Reviewer_3vAw · 2025-11-14
> **Thank you for responses**
>
> I thank the authors for responses to my and (other) reviewer concerns, which overall, I judge quite satisfactory.
>
> However, I am in agreement with Reviewer Z8wQ, in that an updated manuscript showcasing all the suggested revisions would be beneficial.
>
> Additionally, while I find valuable the authors' clarification that the fundamental contribution of the filtration framework is to introduce two new abilities (modeling non-monotonic sequences of graphs and accommodating a more flexible family of graph sequences) and hence, enable the noise augmentation procedure ---critical for the performance boost--- it is still questionable the **direct performance gain** from the actual graph filtration procedure. Namely, it seems that it is the noise augmentation procedure (enabled by the filtration framework) that drives the performance improvements. Precise clarifying statements on this matter would be appreciated.

---

> > ### Author Response · Authors · 2025-11-16
> >
> > We have been working hard on a revised manuscript to incorporate the comments and experiments suggested by all reviewers, including the feedback from the third review that we received recently. We will upload this revision along with our anonymized code as soon as possible.
> >
> > > while I find valuable the authors' clarification that the fundamental contribution of the filtration framework is to introduce two new abilities (modeling non-monotonic sequences of graphs and accommodating a more flexible family of graph sequences) and hence, enable the noise augmentation procedure ---critical for the performance boost--- it is still questionable the direct performance gain from the actual graph filtration procedure. Namely, it seems that it is the noise augmentation procedure (enabled by the filtration framework) that drives the performance improvements. Precise clarifying statements on this matter would be appreciated.
> >
> > We are agree that noise augmentation and the modeling of non-montonic sequence largely drive the performance of ANFM. We also agree that these techniques are enabled by the filtration framework, specifically our choice to model sequences of entire graphs. Below, we further clarify the role the filtration framework. We are updating our revised manuscript to make this more explicit:
> >
> > Existing autoregressive models (e.g., GraphRNN or GRAN) construct graphs by modeling *sequences of node and edge addition operations*. As a consequence, they cannot model non-montonic graph sequences and they are thus not amenable to noise augmentation.
> >
> > We depart from this paradigm by autoregressively modeling sequences of *entire graphs*. Our noise-augmented teacher forcing strategy is based on the following premise: the generative model should generate a sequence of graphs that interpolates between the fully disconnected graph and the target graph while containing both edge addition as well as edge deletion operations. A simple way to construct such sequences for teacher focing is by computing a *monotonic* sequence of subgraphs of the target graph and then applying noise augmentation.
> >
> > Filtrations are a natural and very general way to define these *monotonic* graph sequences. One could certainly restrict the family (e.g., only allowing induced graph sequences) and still perform noise augmentaion and RL finetuning with the methods we have proposed. However, we believe that the more general filtration framework provides additional value and avenues for future research; despite the fact that in our experiments the line Fiedler filtration strategy is less effective than the DFS strategy in many (but not all) of our experiments.
> >
> > To summarize:
> > - Non-monotonic generation and training with noise augmentations are consequences of our choice to model sequences of entire graphs.
> > - Filtration is a flexible and *general* framework for defining monotonic graph sequences to which we can apply noise augmentation.

---

> > > ### Comment · Reviewer_3vAw · 2025-11-26
> > > **Thank you!**
> > >
> > > Dear authors,
> > >
> > > Thank you very much for the updated manuscript and the insightful responses to all reviewers.
> > >
> > > I am in agreement that this revised version provides accurate and convincing evidence of ANFM's performance and significance.
> > >
> > > Just a minor typo on Section 4.5, paragraph "Autoregressive Approaches": "to generate **non-montonic** graph sequences"

---

> > > > ### Author Response · Authors · 2025-11-26
> > > > **Thank you**
> > > >
> > > > Dear Reviewer 3vAw,
> > > >
> > > > Thank you for catching the typo, and we have fixed it. Thank you again for your time and thoughtful review!

---

### Review · Reviewer_Z8wQ · 2025-10-06

**Summary Of Contributions:**

**Summary**

This paper introduces Autoregressive Noisly Filtration Modeling (ANFM), a new method for fast graph generation. Compared to recent diffusion-based graph generators its inference time is significantly shorter, while the graph quality remains comparable. Conceptually, there are three advances: (1) Using general graph filtrations for the autoregressive target sequence instead of relying on node order, (2) noise augmentations during a first teacher-forcing training stage that allow deletions, (3) a discriminator-based free-running second training stage. The latter two aim to mitigate exposure bias of the model during inference. The paper contains extensive experiments and ablations.

**Strengths**
S1: The speed improvement relative to diffusion methods while maintaining decent quality is impressive.
S2: The paper is generally well-written and easy to follow.
S3: Apart from the main model, it is nice that the authors connect filtration-based sequences to classical node order-based sequences and that they comment on the variability of a classical evaluation method due to too few generated samples.
S4: The paper contains several ablations, extensive experiments, clear algorithm descriptions, and a list of used hyperparameters. This helps to judge the importance of individual method components and eases reproducibility.


**Weaknesses**
*Major*
W1: Hyperparameter tweaking per dataset. My main concern with the paper is that several hyperparameters were tweaked for each dataset (Tables 10-12). Were hyperparameters similarly tweaked for competing methods? If not, this raises concerns about the fairness of the comparison.

W2: No clear performance gain from the general graph filtrations by Fiedler vector compared to DFS node-order. While I appreciate the graph-filtration formalism and the explanation that it is a strict generalization of node order-based approaches like DFS, it seems that empirical performance is not clearly better for the line Fiedler vector version. In several cases the DFS version is even better. I think a more detailed discussion and recommendation should be included and the emphasis on graph-filtrations in the abstract, method name, and main contributions should perhaps be reduced.

W3: No code is provided.

*Minor*
W4: Some overselling of results, see W2 and below.

W5: No ablation of MLP Mixer architecture. While the ablations are in general fairly extensive, it would be interesting to see how much performance is due to the choice of the model architecture as opposed to the training process and the filtrations. I.e. keep the two stage-training mode but use a different autoregressive model, e.g., an LSTM.

W6 LSTM performs very strongly in Table 4, but this is not discussed at all.

W7 Uncertainties are only sometimes reported. At least for the very small datasets, it should be fast enough to rerun the methods for several random seeds and report performance variability.

**Questions**
Q1: How does the Fiedler vector of the line graph relate to the Fiedler vector $v$ of the orignal graph and the corresponding filtration by $f(u,v) = max(v_i, v_j)$?
Q2: How exactly are the node-features $v_i^{(t)}$ computed?
Q3: From the complexity analysis it seems like the node-features $v_i^{(t)}$ contain eigenvectors, which makes the complexity of each each step's cubic in the number of nodes. Would it be possible to reuse the previous step's output features $u_i^{(t-1)}$ as $v_i^{(t)}$ instead?

Q4: What is the unit of the reported inference times?

Q5: Why is  ESGG* sometimes slower than DiGress in Table 2, even though it is optimized for inference speed?

Q6: What are the metrics KL-Div and FCD in Table 4?

Q7: The authors write that they removed the exponential moving average from ESGG and found training to be unstable (Appendix G5). Could the missing EMA not be a direct cause for the instability? Also, I do not get why the EMA slows down validation. Only one additional set of weights needs to be stored, right?

Q8: It is a bit unclear how the proposed noise-augmentation of filtrations differs from teacher-forcing with data augmentations. Is the difference simply that a fixed number of augmented sequences are precomputed instead of augmenting on the fly?


**Misc**
Typo "N"-->"T" in proof of Prop B1.

**Audience:**

Yes

**Audience Explanation:**

A fast inference speed is important for using graph generation in practice at scale. The proposed method achieves a significant speed-up while only compromising quality somewhat.

**Claims And Evidence:**

Yes

**Claims Explanation:**

The central claim of fast, decent quality graph generation is well-supported (assuming hyperparameters were not tweaked more for ANFM than for other methods). Smaller claims like the importance of using graph filtrations

> Our approach is fundamentally based on the concept of graph filtration.

are less well supported (see W2).

Moreover, I would recommend a more nuanced discussion of the empirical results. For instance, the authors write

>They [our methods] perform slightly worse than DiGress (Vignac et al., 2023) and ESGG (Bergmeister et al., 2024). While the line Fiedler variant outperforms the DFS variant on the planar graph dataset, the DFS variant performs substantially better on the SBM dataset.

But the VUN score of ANFM is much, not slightly, worse than that of ESGG on the planar dataset in Table 1. Similarly the line Fiedler variant is worse in all MMD metrics on the planar graph dataset, but much better in VUN. So it neither out- nor underperforms compared to the DFS variant.

**Requested Changes:**

In addition to the weaknesses above:

C1: Include some illustration of the Fiedler vector filtration in the main paper (like Fig 6 a, b)

C2: Please indicate best (/ second best) performing methods in bold (/ italics) in Tables 1-4 etc like it is done in Tables 20-26.

C3: Please mention the default number of diffusion steps of DiGress in the caption of Fig 2 or extend the x-axis (logarithmically) to this default number.

C4: Given the linear complexity of ESGG vs the cubic complexity of ANFM, it would be helpful to get a feeling for the size of graphs at which ESGG really becomes much faster. E.g. one could take an ESGG and an ANFM model trained on the small graphs from the paper and then ask these models to generate progressively larger graphs tracking the inference time.

C5: I appreciate the discussion on evaluation variability in I.1. It would become even stronger if the authors included a table similar to Table 27, but directly for the validity estimates, showing that the std goes down with the number of model samples (but I expect the mean not decrease consistently as for MMD in Table 27, since even for a small number of samples the estimator is unbiased).

---

> ### Author Response · Authors · 2025-10-18
> **Response to Reviewer Z8wQ**
>
> **Edit: We posted our response on Oct. 18, and just realized that our response is not visible to you. We are really sorry for any inconvenience.**
>
> Thank you for your time and effort in providing valuable feedback! We appreciate your recognition of our work's strengths and your constructive suggestions for improvement. Below, we address each of your concerns and questions, and we will update our manuscript as soon as we receive all reviews.
>
> ### W1: Hyperparameter tweaking per dataset
>
> This is an important concern regarding fairness. We want to clarify that **all baseline methods received comparable hyperparameter tuning efforts.** Specifically:
>
> - For DiGress and ESGG, we used the official implementations with the hyperparameters (learning rates, hidden dimensions, number of layers) reported in the original work, which were carefully selected per dataset (for example, [DiGress official codebase](https://github.com/cvignac/DiGress/tree/main/configs/experiment) has a specific config for each considered dataset).
> - For GRAN, we adapted the publicly available implementations, used the officially published checkpoint for the proteins dataset, and performed careful model selection for the other datasets (c.f., Appendix G). We ensured that the performance we observed on the synthetic datasets was comparable to the performance reported in previous works.
>
> We hope this demonstrates that our comparisons are fair and that ANFM's performance gains are not simply due to more extensive hyperparameter optimization.
>
> ### W2: No clear performance gain from the general graph filtrations by Fiedler vector compared to DFS
>
> We appreciate this critical observation and agree that it deserves more prominent discussion. You are correct that DFS filtrations often perform comparably or better than line Fiedler filtrations, particularly on the SBM dataset.
>
> However, we want to emphasize a crucial aspect: **the filtration framework's primary contribution is enabling non-monotonic sequence modeling, not necessarily the use of non-induced subgraphs (like line Fiedler filtrations).** This distinction is important because:
>
> - **Non-monotonic modeling is a prerequisite for noise augmentation:** Traditional node-addition approaches (GraphRNN, GRAN) can only generate monotonic sequences and thus cannot leverage noise augmentation, which our ablations show is a major driver of performance.
> - **DFS filtrations within our framework already benefit from this capability:** Even the DFS variant benefits from the filtration formalism because it enables the two-stage training procedure with noise augmentation.
> - **Our filtration framework provides more flexibility:** our filtration framework enables modeling arbitrary sequences through the filtration function and schedule. While our work studied a few of them, we believe it provides new insights into the choice of the graph sequence for autoregressive modeling.
>
> **Proposed manuscript changes:** We will add a prominent discussion in the results section acknowledging that DFS filtrations often perform better in practice and we will clarify that the generalization to arbitrary graph filtrations is a theoretical contribution that enables future exploration, while the practical benefit primarily comes from non-monotonic modeling.
>
> ### W3: No code provided
> We are fully committed to open and reproducible research. **We have prepared our code for publication and will release it upon acceptance of the paper.** We will add a statement to the manuscript confirming this commitment.
>
> ### Minor weakness
>
> - W4: Thank you for pointing this out. We agree that our language should be more precise, and we will update our manuscript accordingly.
> - W5: Thank you for your suggestion. While we agree with the reviewer that comparing an LSTM and a Transformer decoder in our backbone architecture would be interesting, we believe this experiment is not necessary to justify our choice. In many other fields, like NLP, the transformer has been proven to be more powerful than LSTM in practice. The initial goal of our work is to provide a model architecture that can model our filtration sequences. We leave the systematic study of different components in our model architecture for future work.
> - W6: You're absolutely right that this deserves discussion. The LSTM model in Table 4 is an LSTM model trained on SMILES strings, a commonly used baseline. We will include a discussion in our revised manuscript.

---

> ### Author Response · Authors · 2025-10-18
> **Response to Reviewer Z8wQ (cont.)**
>
> W7: We appreciate this concern and acknowledge the importance of reporting uncertainties. We recall that we report uncertainty estimates across three training runs of ANFM for all expanded synthetic datasets in Appendix F.1. However, we would like to respectfully clarify some practical considerations:
> - While the small synthetic datasets contain relatively few graphs, training models on them is not computationally trivial. For instance, training a single DiGress or ESGG model on the small SBM dataset requires approximately one week of compute time on high-end GPUs. Running multiple seeds for all baselines across all datasets would require prohibitive computational resources (several months of GPU time).
> - Importantly, we are among the first to explicitly identify and address the robustness issues arising from evaluating on small datasets. Our introduction of the **expanded synthetic datasets** directly mitigates this limitation and enables more stable performance estimation. On these expanded datasets, we observe much more consistent results that better reflect true model performance.
>
> ## Requested Changes
>
> We will update our manuscript to take into account your requested changes C1-C3.
>
> ### C4 Inference time scaling with graph size
>
> We appreciate this suggestion, but have some concerns about its feasibility and interpretation.
>
> 1. Models trained on small graphs may generate unrealistic intermediate structures when asked to produce larger graphs, leading to inference costs that do not reflect realistic use cases.
> 2. Out-of-distribution behavior could lead either model to generate unusually sparse or dense graphs, unfairly skewing the results.
> 3. Fair comparison would require retraining both models with architectures scaled appropriately for larger graphs, substantially expanding the scope of the experiment.
>
> Instead, **we propose adding a brief discussion** of how inference costs scale with graph size for both methods in the main text, along with highlighting empirical measurements on the largest graphs in our datasets (i.e., the Proteins dataset).
>
> ### C5: Validity estimate variability analysis
>
> Thank you for appreciating our analysis on evaluation variability. We believe that these issues are a significant hurdle for the field.
>
> As you correctly note, the validity metric (being a binomial proportion) has a closed-form standard deviation given by Proposition I.1, which is why we did not include a table. However, **we agree that adding a table would strengthen the presentation** by making the analysis more concrete and accessible. We are happy to add a table with standard deviations computed via this formula.

---

> ### Author Response · Authors · 2025-10-25
> **Answering Questions of Reviewer Z8wQ**
>
> Below, we answer your questions.
>
> > Q1: How does the Fiedler vector of the line graph relate to the Fiedler vector of the orignal graph and the corresponding filtration
>
> Let $G$ be a graph with $n$ nodes and $m$ edges. The Fiedler vector of $G$ is an $n$-dimensional vector while the Fiedler vector of the line graph $L(G)$ is $m$-dimensional. The Fiedler vector of $G$ operates on nodes and captures node-level connectivity, while the Fiedler vector of $L(G)$ operates on edges and captures edge-level connectivity patterns.
>
> The filtration based on the Fiedler vector of L(G) tends to add edges in an order that respects the global edge connectivity structure, which can differ substantially from node-based orderings.
> We are not aware of any specific relationship between the two vectors and their corresponding filtrations.
>
> > Q2: How exactly are the node-features computed?
>
> We provide details about the computation of node features in appendix E.1. We are happy to provide further details if needed.
>
> > Q3: From the complexity analysis it seems like the node-features contain eigenvectors, which makes the complexity of each each step's cubic in the number of nodes. Would it be possible to reuse the previous step's output features instead?
>
> We believe that using the output features $u$ of one step as the input features $v$ of the next step would be problematic for the following two reasons:
>
> - Effectively, no Laplacian eigenvectors would be computed as the eigenvectors of the Laplacian of the first graph are trivial. We expect that this would limit the expressivity of our model.
> - This approach would make the architecture recurrent, requiring unrolling and backpropagation through time during training, which may be expensive and unstable. In contrast, our current architecture uses a transformer decoder to incorporate information from previous timesteps. This allows the model to be trained in a single efficient forward and backward pass without unrolling.
>
> > Q4: What is the unit of the reported inference times?
>
> The unit is "seconds per graph". We compute these values by measuring the time required to generate 1024 graphs and dividing that value by 1024.
>
> > Q5: Why is ESGG* sometimes slower than DiGress in Table 2, even though it is optimized for inference speed?
>
> ESGG is designed to scale well to large graphs, achieving a favorable *asymptotic* complexity compared to DiGress. However, this improvement only becomes noticeable at relatively large graph sizes. This can also be observed in Figure 6 in the appendix of the ESGG paper: ESGG is slower than DiGress for graphs containing roughly fewer than 500 nodes.
>
> > Q6: What are the metrics KL-Div and FCD in Table 4?
>
> These are metrics that measure the sample quality of valid generated molecules:
> - FCD denotes the Fréchet ChemNet distance. Molecules are converted to SMILES strings and then embedded via a pre-trained LSTM model. The similarity of generated and reference molecules is assessed by comparing these embeddings, analogous to the popular Fréchet Inception distance (FID), in image generation.
> - For the KL-Div metric, physicochemical parameters of generated and reference molecules are computed. The distributions of these parameters are compared via their KL divergence.
>
> In the context of the GuacaMol benchmark, a scaled negative exponential of the distance/divergence metrics is reported. This maps these metrics to the unit interval. For more details, we refer to Brown et al, 2019. We will update our manuscript to discuss these metrics in more detail.
>
> > Q7: The authors write that they removed the exponential moving average from ESGG and found training to be unstable (Appendix G5). Could the missing EMA not be a direct cause for the instability? Also, I do not get why the EMA slows down validation. Only one additional set of weights needs to be stored, right?
>
> During validation, graphs have to be sampled from all EMA models independently. Since inference is expensive for ESGG (c.f., Table 2) and validation has to be performed at many checkpoints for proper model selection, we found that including many EMA variant slowed training substantially. Indeed, more time was spent in validation than the actual training loop.
>
> In preliminary experiments we did not find EMA variants to perform substantially better than the non-smoothed variant and we deemed the evaluation of EMA models computationally infeasible.

---

> ### Author Response · Authors · 2025-10-25
> **Answering Questions of Reviewer Z8wQ (cont.)**
>
> > Q8: It is a bit unclear how the proposed noise-augmentation of filtrations differs from teacher-forcing with data augmentations. Is the difference simply that a fixed number of augmented sequences are precomputed instead of augmenting on the fly?
>
> We agree with the reviewer that our first training stage constitutes teacher-forcing on noise-augmented filtration sequences.
>
> In principle, augmenting filtration sequences on the fly is possible. However, due to the required eigen decompositions, we found that this slowed down training to some degree. Hence, we decided to pre-compute augmentations. Please note that traditional autoregressive models such as GraphRNN and GRAN fundamentally cannot model non-montonic sequences, making noise augmentation inapplicable without our filtration formulation.
>
> We are unsure whether we correctly understood the intent of your question. We are happy to further discuss this point if our answer does not address your concern.

---

> > ### Comment · Reviewer_Z8wQ · 2025-11-13
> >
> > I thank the authors for their detailed reply. My main issue about tweaking hyperparameters per dataset has been resolved. Apparently this is standard practice and was performed for all compared methods, ensuring fairness. While the authors have promised to address several of my concerns in the revision, I wonder why they did not address them already and upload a revised manuscript, so that the reviewers can check the proposed changes. This would strengthen the rebuttal a lot!
> >
> > On some points, I still have questions:
> >
> > > W2 and Q8: Merit of the filtration framework
> >
> > I did not fully understand how the filtration framework allows for non-monotonic modeling / noise augmentation. Any graph filtration is, by definition, a sequence of subgraphs, so monotonic, right? Thus, while performing noise augmentations and non-monotonic modeling seems to be a good idea, it is not an upshot of the filtration framework. In particular, a noise-augmented, non-monotonic training sequence will *not* be a graph filtration anymore. Could the authors please clarify this point?
> >
> > If the two contributions, general filtrations and noise augmentations, are really separate, my point about limited practical usefulness of the filtration framework stands (while it might still be theoretically interesting).
> >
> >
> > > Q2: Node features
> >
> > Appendix E.1 mentions
> > > Laplacian positional encodings (Dwivedi et al., 2023), random walk positional encodings (Dwivedi et al., 2022), and cycle counts following DiGress
> >
> > Would it be feasible to reproduce the definitions of these in the appendix for more self-contained read?

---

> > > ### Author Response · Authors · 2025-11-16
> > >
> > > We have been working hard on a revised manuscript to incorporate the comments and experiments suggested by all reviewers, including the feedback from the third review that we received recently. We will upload this revision along with our anonymized code as soon as possible.
> > >
> > > > Any graph filtration is, by definition, a sequence of subgraphs, so monotonic, right? Thus, while performing noise augmentations and non-monotonic modeling seems to be a good idea, it is not an upshot of the filtration framework. In particular, a noise-augmented, non-monotonic training sequence will not be a graph filtration anymore.
> > >
> > > Existing autoregressive models (e.g., GraphRNN or GRAN) construct graphs by modeling *sequences of node and edge addition operations*. As a consequence, they cannot model non-montonic graph sequences and they are thus not amenable to noise augmentation.
> > >
> > > We depart from this paradigm by autoregressively modeling sequences of *entire graphs*. Our noise-augmented teacher forcing strategy is based on the following premise: the generative model should generate a sequence of graphs that interpolates between the fully disconnected graph and the target graph while containing both edge addition as well as edge deletion operations. A simple way to construct such sequences for teacher focing is by computing a *monotonic* sequence of subgraphs of the target graph and then applying noise augmentation.
> > >
> > > Filtrations are a natural and very general way to define these *monotonic* graph sequences. One could certainly restrict the family (e.g., only allowing induced graph sequences) and still perform noise augmentaion and RL finetuning with the methods we have proposed. However, we believe that the more general filtration framework provides additional value and avenues for future research; despite the fact that in our experiments the line Fiedler filtration strategy is less effective than the DFS strategy in many (but not all) of our experiments.
> > >
> > > To summarize:
> > > - Non-monotonic generation and training with noise augmentations are consequences of our choice to model sequences of entire graphs.
> > > - Filtration is a flexible and *general* framework for defining monotonic graph sequences to which we can apply noise augmentation.
> > >
> > > In this sense the two contributions are not separate since noise augmentation requires sequences of entire graphs constructed by the filtration.
> > >
> > >
> > > > Would it be feasible to reproduce the definitions of these in the appendix for more self-contained read?
> > >
> > > We will add a description of the positional and structural encodings to our revision.

---

> > > > ### Comment · Reviewer_Z8wQ · 2025-11-25
> > > >
> > > > I thank the authors for the additional reply and, in particular, for providing a revised version of the manuscript. I think the performance contributions of the non-monotonic noise augmentations vs the general graph filtration are now appropriately described. I recommend acceptance.

---

> > > > > ### Author Response · Authors · 2025-11-26
> > > > > **Thank you**
> > > > >
> > > > > Dear Reviewer Z8wQ,
> > > > >
> > > > > Thank you again for your time and thoughtful review!

---

### Review · Reviewer_tqkU · 2025-11-05

**Summary Of Contributions:**

Summary:
The paper presents Autoregressive Noisy Filtration Modeling (ANFM), an edge autoregressive framework for graph generation. During training, a filter function determines the order in which edges are gradually masked, working as a schedule. The generative model leverages a GNN that processes intermediate graph structures, and a transformer decoder to handle the temporal axis. Sampling is performed via mixtures of multivariate Bernoulli distributions. The model is shown to perform comparably to the state of the art while significantly reducing generation time.

Strengths:
1. The paper is generally well written, without typos (as far as I noticed), and easy to follow.
2. The idea of using filtration to model the graph edge "editing" is interesting. The introduction of additional techniques, namely the use of noise and adversarial finetuning to counter exposure bias, is well motivated.
3. The model is evaluated on many datasets from the graph generation literature, and I appreciated the introduction of at least one dataset with labels.

Weaknesses:
1. Some parts are missing motivation/explanation. See questions for details.
2. The choice of the median to report results (across 3 runs) instead of the mean is a strange and not well-motivated choice. It could be in good faith, but using the median could allow for reporting higher results when only 2 of the runs are good, while 1 is bad. Additionally, I did not understand whether the maximum deviance is with respect to the mean or the median.
3. Smaller issue: the code is missing. If possible and available, I would recommend sharing the code through anonymized github, or other means.

Questions regarding Weakness 1:
1. What is the motivation behind the Line Fiedler Filtrations, apart from allowing non-induced subgraphs? Can you provide a more thorough explanation and motivation of the method, and, if possible, your opinion on different results with respect to Depth First Search?
2. What is the KL Div in Table 4, and why is it the higher the better? Same for FCD.
3. Why did you use the mixture of multivariate Bernoulli distributions instead of a single categorical/bernoulli, as is done in, e.g., DiGress?

General questions:
4. Why was the median used instead of the mean?
5. I'm interested in your opinion on the results of Transformer vs. First-Order, where the fully autoregressive variant performed better. Shouldn't all the information already be stored in \tilde{G}_t? Does it have to do with the filtration strategy?
6. Is the GNN (implemented as a Graph Transformer) propagating information in a fully connected fashion, or only through connected nodes? In the former case, is it still true that ANFM can generate sequences of induced subgraphs? (because nodes already existing can influence the generative process)
7. Does the use of GAN finetuning on the last step influence all the features extracted throughout the generative sequence (by backprop through the transformer)?

**Audience:**

Yes

**Audience Explanation:**

The topic of graph generation, especially with the aim of high throughput, is very important.

**Broader Impact Concerns:**

There are no concerns or ethical implications of the work.

**Claims And Evidence:**

No

**Claims Explanation:**

I did not find an explanation for the use of median instead of the mean (on 3 runs). Although in good faith, the use of median could hide a potentially bad run out of the 3. See Questions for more details. I will be happy to change my mind if an explanation on the use of median, or a switch to the mean (or alternatives) is done.

**Requested Changes:**

Critical changes:
1. Including a motivation for the use of median, or a switch to the mean
2. Inclusion of the topics touched on in Q1-4

Minor changes:
1. Inclusion of the topics touched on in Q5-7
2. Inclusion of the code

---

> ### Author Response · Authors · 2025-11-11
> **Response to Reviewer tqkU**
>
> Thank you for your time and effort in providing valuable feedback!
> We appreciate your recognition of our work's strengths and your constructive suggestions for improvement. Below, we address each of your concerns and questions. We will include all the suggested changes in our revised manuscript.
>
> ### Weaknesses
>
> > I did not understand whether the maximum deviance is with respect to the mean or the median.
>
> The maximum deviation is reported w.r.t. the median. Concretely, we perform three training runs and compute corresponding performance metrics $x_1 \leq x_2 \leq x_3$. We report the median as $m := x_2$ and the maximum deviation as $d := \max(|x_1 - x_2|, |x_3 - x_2|)$.
>
> Hence, from our reported numbers, you may reconstruct a lower bound on the "worst-case" scenario as $m - d$ (assuming a metric that should be maximized, e.g. VUN). Overall, you may infer the following:
> - All of the three runs have achieved **at least the performance $m - d$**.
> - Two of the three runs achieved at least the performance $m$.
>
> > The choice of the median to report results [...] instead of the mean is a strange and not well-motivated choice. It could be in good faith, but using the median could allow for reporting higher results when only 2 of the runs are good, while 1 is bad.
>
> The choice of evaluation methodology was made in good faith ex-ante, prior to any computational experiments. We are still convinced that the median with maximum deviation is an appropriate quantification of expected performance and variability:
>
> - Due to a constrained computational budget, we could only perform three training runs per dataset. It is an accepted rule of thumb that empirical standard deviations may only be reliably computed with at least 10 trials. We hence ruled out standard deviations as a quantification of variability and instead chose the maximum deviation.
> - As outlined in the previous response, our reported numbers allow the reader to infer information about all three runs: **The median minus the maximum deviation provides a lower bound on the performance of the worst run**. Additionally, the median itself lower bounds the performance of the other two runs. **In contrast, reporting the mean would not allow such reasoning over the performance of all three runs**.
>
> In practice, **we do not encounter the case where one of the three runs fails and the median masks this issue**. Below, we show the VUN metrics across all three model runs of the Line Fiedler ANFM model as well as the mean and median (note that we did not quantify uncertainties for the DFS variant).
> For all datasets, mean and median provide similar estimates and for two of the three datasets, the median (which we report) provides a more conservative performance metric.
>
> VUN for Line Fiedler ANFM:
> | Dataset | Run A | Run B | Run C | Mean | Median |
> |---------|-------|-------|-------|------|--------|
> Expanded Planar | 0.7919921875 | 0.86328125 | 0.7724609375 | **0.8092** | 0.792 |
> Expanded SBM  | 0.759765625 | 0.7451171875 | 0.796875 | **0.7673** | 0.7598|
> Expanded Lobster  | 0.8037109375 | 0.791015625 | 0.7197265625 | 0.7715 | **0.791** |
>
> We are committed to reproducible research and thank the reviewer for this important question. All pretrained models will be released upon publication, and we are happy to provide further analysis and additional results if the reviewer needs any further clarification.
>
> > the code is missing
>
> We have prepared the code for publication. We will anonymize it and provide it as supplementary material.
>
> ### Further Questions
>
> > Q1: What is the motivation behind the Line Fiedler Filtrations
>
> We were motivated by the intuition that it may be beneficial to construct graphs in a local fashion, i.e., adding edges that are incident to the same vertex at similar timesteps in the generation process. The line Fiedler filtration achieves this since the Fiedler vector of the line graph paints the edges in a continuous gradient (c.f. Figure 6a).
>
> However, our experiments showed that the DFS variant, which does not possess this property (c.f. Figure 6b), could perform better in some cases. As we introduced our method as a general framework, we believe including all these variants would be helpful for practitioners.

---

> > ### Author Response · Authors · 2025-11-11
> > **Response to Reviewer tqkU (cont.)**
> >
> > > Q2: What is the KL Div in Table 4, and why is it the higher the better? Same for FCD.
> >
> > We strictly follow the original benchmarking tool provided by the GuacaMol benchmark. These metrics measure the sample quality of valid generated molecules (Brown et al, 2019). Specifically,
> > - FCD denotes the Fréchet ChemNet distance. Molecules are converted to SMILES strings and then embedded via a pre-trained LSTM model. The similarity of generated and reference molecules is assessed by comparing these embeddings, analogous to the popular Fréchet Inception distance (FID), in image generation.
> > - For the KL-Div metric, physicochemical parameters of generated and reference molecules are computed. The distributions of these parameters are compared via their KL divergence.
> >
> > In the GuacaMol benchmark, a **scaled negative exponential** of the distance/divergence metrics is reported. As a consequence, larger values indicate better performance. For more details, we refer to Brown et al, 2019. We will update our manuscript to discuss these metrics in more detail.
> >
> > > Q3: Why did you use the mixture of multivariate Bernoulli distributions instead of a single categorical/Bernoulli, as is done in, e.g., DiGress?
> >
> > We were motivated by the fact that the probability $p(\tilde G_t | \tilde G_{t - 1}, ..., \tilde G_0)$ can, a priori, not be decomposed as a product over the edges. In other words, the random variables indicating edge existence in $\tilde G_t$ are not independent. We aim to model these dependencies through the mixture distribution in $p_\theta$. A similar approach has been employed in the autoregressive GRAN model (Liao et al, 2019).
> >
> > > Q4: Why was the median used instead of the mean?
> >
> > See our detailed answer above in the weakness section.
> >
> > > Q5:  I'm interested in your opinion on the results of Transformer vs. First-Order, where the fully autoregressive variant performed better. Shouldn't all the information already be stored in \tilde{G}_t? Does it have to do with the filtration strategy?
> >
> > For the sake of this discussion, we only consider the first training stage, i.e., teacher forcing.
> >
> > The model $p_\theta$ is trained to model the posterior $p(\tilde G_{t+1} | \tilde G_{t}, ..., \tilde G_0)$. Since the forward process $\tilde G_0, ..., \tilde G_T$ is *not necessarily Markov* (due to our filtration formulation), neither is the reverse process $\tilde G_T, ..., \tilde G_0$. Hence, we may have $p(\tilde G_{t+1} | \tilde G_t, ..., \tilde G_0) \neq p(\tilde G_{t+1} | \tilde G_{t})$. **In this sense, not all information is contained in $\tilde{G}_t$**. This contrasts ANFM against diffusion models where *both the forward and reverse processes are Markov* and a first-order structure is hence sufficient.
> >
> > Nevertheless, we note that the first-order variant of ANFM may remain a somewhat competitive alternative to the fully autoregressive one after GAN tuning (c.f. Table 9).
> >
> > > Q6: Is the GNN (implemented as a Graph Transformer) propagating information in a fully connected fashion, or only through connected nodes? In the former case, is it still true that ANFM can generate sequences of induced subgraphs?
> >
> > We use SAT, a graph transformer architecture, which includes both message passing layers and global self-attention across all nodes. Hence, information also propagates between distinct connected components.
> >
> > We do not fully understand the second part of your question. It would be helpful if you could clarify it.
> >
> > > Q7: Does the use of GAN finetuning on the last step influence all the features extracted throughout the generative sequence (by backprop through the transformer)?
> >
> > This is correct. While we use only the output of the last generative step for the reward calculation, gradients flow through all generation steps.
> >
> > Specifically, in PPO, we compute the reward-to-go at every generation step (which is identical to the terminal reward), perform advantage estimation at every step, and backpropagate the PPO objective at every step.

---

> > > ### Comment · Reviewer_tqkU · 2025-11-18
> > >
> > > Dear Authors,
> > >
> > > Sorry for the late response. Thank you for the clarifications, particularly regarding the use of the median to report results. I understand and agree with your reasoning behind it, and appreciate that you added an explanation in the paper.
> > >
> > > ## About the Line Fiedler Filtration
> > > Although I understood the "shallow" why, i.e., to have non-induced subgraphs, I still don't understand the "deeper" why: why did you construct this filtration function the way you did (mathematically)? What is the role of the Fiedler vector in the Laplacian of the line graph? How is f defined for the Fiedler filtration? (e.g., for DFS you provided a formal definition) Are there references that may allow a reader to improve its understanding of this technique (e.g., to the Fiedler vector)? I think clearly presenting how this filtration works would enhance the paper, and make motivating this technique easier.
> > >
> > > Finally, there is a typo after Eq.5, Fielder -> Fiedler
> > >
> > > ## About Transformer vs. First-Order
> > > Thanks for the interesting discussion, it actually makes a lot of sense.
> > >
> > > ## About ANFM generating induced subgraphs
> > > To clarify the point, let me clearly state the definition of an induced subgraph (to be on the same page):
> > >
> > > Let G=(V,E) be a graph with vertices V and edges E\subseteq V\times V. The graph G_I=(V', E') is an induced subgraph of G if V'\subseteq V and E'=E\cap V'\times V'.
> > >
> > > Now, The graph G_M=(V,E'), i.e., with the edges of the induced subgraph and vertices of the original graph, is not an induced subgraph of G. This would be a minor of graph G. From your definition of filtration (eq. 1) I can assume that you are operating with minors G_M, and not induced subgraphs G_I.
> > >
> > > Ideally, ANFM does not align with the same problem solved via node addition, as it depends on the information available during generation. For example, considering a graph $G_t$ at time $t$, autoregressive models don't have access to the final number of nodes to generate. This inductive bias allows them to learn to extend graphs (e.g., G_1 may be obtained starting from G_2, with G_2 induced subgraph of G_1). On the other hand, ANFM has access to the number of final nodes during the generative process, due to the global self attention. Then G_1 cannot be obtained from G_2 with a lower number of nodes (also requiring to have feature vectors for the added nodes).
> > >
> > > ## About the gradient flow
> > > Thank you for the clarifications.
> > >
> > >
> > > You can consider all the points I did not mention here as solved.
> > > Be careful though about the size: I checked the revised paper, and even removing the dashed text it doesn't look like it will fit the 12 pages of main content.

---

> > > > ### Author Response · Authors · 2025-11-19
> > > >
> > > > > Although I understood the "shallow" why, i.e., to have non-induced subgraphs, I don't understand the "deeper" why: why did you construct this filtration function the way you did (mathematically)? What is the role of the Fiedler vector in the Laplacian of the line graph?
> > > >
> > > > We believe that there may be a slight misunderstanding regarding our intentions in studying the line Fiedler filtrations: while this filtration strategy does represent one choice that yields graphs with edge sets that do not correspond to induced subgraphs, this was not our primary goal.
> > > >
> > > > Rather, we aimed to design an edge weight function that, loosely speaking, paints the edges of a given graph in a continuous gradient. Intuitively, we expect this to be beneficial because it allows graphs to be constructed in a local fashion (i.e., adding edges iteratively in similar regions). The line Fiedler filtration function achieves exactly this goal. We refer to Figure 1a which illustrates its behavior on a planar graph. One can appreciate how the line Fiedler function paints the edges in a smooth gradient from one end of the graph to another.
> > > >
> > > > We will now elaborate on why, mathematically speaking, we chose the Fiedler vector of the line graph to achieve the goal above. It is well known that the Fiedler vector of a given graph $G = (V, E)$ (i.e., the first non-trivial eigenvector of the symetrically normalized graph Laplacian) assigns similar values to neighboring vertices (a fact that is famously leveraged in spectral clustering). Hence, the *vertices* of G are colored in a smooth gradient. Since we intend to color the *edges* of G in a smooth gradient, we transition to the line graph of $G$, denoted $L(G)$. Edges of $G$ form vertices in $L(G)$ and they are adjacent in $L(G)$ if they are incident to a shared vertex in $G$. The Fiedler vector of $L(G)$ may now be seen as an edge weight function on $G$. By the previously discussed property of Fiedler vectors, it will assign similar values to edges that are adjacent in $L(G)$, i.e., edges that share an incident vertex in $G$. This results in a "smooth" edge weight function on $G$.
> > > >
> > > > > How is f defined for the Fiedler filtration? (e.g., for DFS you provided a formal definition)
> > > >
> > > > Unfortunately, we cannot give a concise formula for the line Fiedler function since it is defined as the first non-trivial (i.e., the second smallest) Laplacian eigenvector of the line graph, which does not have a closed expression based on the graph. In particular, this eigenvector is at most unique up to its sign. However, we will provide a more in-depth definition in the appendix, consistent with our mathematical explanation above.
> > > >
> > > > > Now, The graph G_M=(V,E'), i.e., with the edges of the induced subgraph and vertices of the original graph, is not an induced subgraph of G. This would be a minor of graph G. From your definition of filtration (eq. 1) I can assume that you are operating with minors G_M, and not induced subgraphs G_I.
> > > >
> > > > We agree on this point. We note this subtle distinction between induced subgraphs and the subgraphs obtained in the DFS filtration in our methods section: "Hence, this filtration closely mirrors the sequence of induced subgraphs, differing only in that the node set does not change over time."
> > > >
> > > > For the same reason, we also agree that, mathematically speaking, ANFM does not model sequences of induced subgraphs. However, it may be trained on filtrations that consist of induced subgraphs and additional isolated vertices. In this sense, it may still mirror node-addition approaches.
> > > >
> > > > We will revise our language to more precisely differentiate between induced subgraphs and subgraphs that additionally contain isolated vertices.
> > > >
> > > >
> > > > > Be careful though about the size: I checked the revised paper, and even removing the dashed text it doesn't look like it will fit the 12 pages of main content.
> > > >
> > > > We have reached out to the action editor to inquire on the page limit policy for revisions. If it needs to be reduced to 12 pages we will do so as soon as possible.

---

> > > > > ### Comment · Reviewer_tqkU · 2025-11-21
> > > > >
> > > > > Dear Authors,
> > > > >
> > > > > > We will now elaborate on why, mathematically speaking, we chose the Fiedler vector of the line graph to achieve the goal above.
> > > > >
> > > > > Thank you for the mathematical definition, this was exactly the explanation I was looking for. In particular you answered: what value is assigned to each edge (the definition of $f(e)$), why it matters (the values of this $f(e)$ form a smooth gradient on the edges), and what is the relation with the Laplacian of the line graph (to compute spectral clustering on the edges).
> > > > >
> > > > > In my opinion, many of these concepts that made me understand the technique were not present in the manuscript, and I was also not aware of some facts you mentioned. I think the presentation of this technique would be greatly improved by well-defining $f(e)$ (as you already did for DFS), and following a line of reasoning without gray areas (e.g., why use the Fiedler vector of the line graph, mathematically speaking), including some references. I would suggest having the explanation in the introduction of the Line Fiedler Filtration due to its importance. These additions might also make it easier to motivate the technique.
> > > > >
> > > > > > Unfortunately, we cannot give a concise formula for the line Fiedler function since it is defined as the first non-trivial (i.e., the second smallest) Laplacian eigenvector of the line graph, which does not have a closed expression based on the graph.
> > > > >
> > > > > I agree with this. In fact, I was not looking for the definition of the Fiedler vector, but of the filtration function, which is mentioned in the paper only as:
> > > > > > This function is derived from the second smallest eigenvector (Fiedler vector)
> > > > >
> > > > > Let me know what you think.

---

> > > > > > ### Author Response · Authors · 2025-11-24
> > > > > > **Revision to improve presentation of the line Fiedler filtration strategy**
> > > > > >
> > > > > > We agree that clarity can be improved by more clearly tracing out our line of reasoning in the main paper. We have updated our manuscript accordingly, marking the new changes in green.

---

> > > > > > > ### Comment · Reviewer_tqkU · 2025-11-25
> > > > > > >
> > > > > > > Thank you. I have no further questions and, after the fruitful dialogue with the authors, I recommend acceptance.

---

> > > > > > > > ### Author Response · Authors · 2025-11-26
> > > > > > > > **Thank you**
> > > > > > > >
> > > > > > > > Dear Reviewer tqkU,
> > > > > > > >
> > > > > > > > Thank you again for your time and thoughtful review!

---

### Author Response · Authors · 2025-11-17
**Paper revision**

We thank all reviewers again for their valuable feedback, which we believe has strengthened our paper. We have incorporated all their suggestions into our revised manuscript. We have uploaded this revision and additionally provided our anonymized code as supplementary material, which we will release on GitHub once the paper is published.

### Summary of the revision
We have highlighted additions and modifications to our manuscript in red and deletions in blue strikethrough. Below, we summarize a non-comprehensive list of the most important changes.

- **Clarifying the contribution of the filtration framework**: Reviewers have raised questions about the contribution of the filtration framework to the performance of ANFM. We aim to clarify that the performance of ANFM is largely driven by its ability to model non-montonic graph sequences, which is incentivized via RL finetuning and noise augmentations and enabled by our filtration framework. To this end:
    - We have removed a sentence from the abstract stating that ANFM "generalizes prior autoregressive methods by enabling the construction of non-induced subgraphs". While we maintain this claim, we are striking it from the abstract to shift focus to the modeling of non-monotonic graphs, the key driver of performance.
    - We have expanded Section 4.5 to elaborate on which sense the filtration framework facilitates the modeling of non-montonic sequences (and consequently noise augmentation).
    - We have updated our experiments section 5.2 to explicitly state that the DFS variant outperforms the line Fiedler variant on two of the three datasets.
    - We have updated our conclusion to prominently note that (1) noise augmentation and RL finetuning are the main drivers of performance and (2) the DFS filtration strategy oftentimes outperforms the line Fiedler strategy.
- **Extension of the experiments on filtration granularity and inference costs**
    -  We have trained an ANFM model on the expanded planar graph dataset using 60 steps and have updated Figure 2 to include the results. We observe that generation quality remains similar to the quality at 30 and 45 steps, while the inference time continues to increase in a roughly linear fashion.
    -  In Appendix L we have analyzed how the MMD changes as the number of generation steps varies in ANFM and DiGress. The findings are consistent with those from Figure 2.
- Analysis of the training costs: We have compared the training costs of ANFM to the training costs of DiGress in Appendix K.
- We have justified why we use a fixed $T$ for all graphs in Section 4.1.
- We have added a remark that existing graph generators may be amenable to RL finetuning, but that this requires non-trivial modifications and thus falls outside the scope of our work in Section 4.4.
- We have included illustrations of filtration functions in Figure 1.
- We have indicated the best and second-best performing methods in bold and underlined fonts across all tables.
- We have provided a definition of the FCD and KL Div metrics in Section 5.3.
- We have justified our choice of reporting median performance with maximum deviations to quantify uncertainties in Appendix F.1
- We have provided our motivation for studying the line Fiedler filtration function in Section 4.1.
- We have elaborated on the reasons for using mixtures of multivariate Bernoulli distributions as opposed to simple multivariate Bernoulli distributions in Section 4.3.

---

### Author Response · Authors · 2025-11-19
**Minor revision to make language more precise**

Dear reviewers,

following feedback from one reviewer, we have made minor modifications to make our language more precise: when we talk about subgraphs constructed in the DFS filtration strategy, we refer to them as "having induced edge sets" and not as them "being induced". This language is more precise since the subgraphs constructed in the DFS filtration consist of an induced subgraph and *isolated vertices* (i.e., they have the edge set of an induced subgraph but all vertices of the original graph).

While we have originally noted this distinction in our methods section ("... yields a
filtration sequence in which the edge set of Gt coincides with the edge set of G[Vt] for any t. Hence, this filtration closely mirrors the sequence of induced subgraphs, differing only in that the node set does not change over time. "), our language was less precise in the rest of the paper and particularly in the revision.

We hope that these new changes improve clarity. They are marked in cyan.

---

### Decision · Action_Editor_Eihc · 2026-01-08

**Recommendation:** Accept as is

**Additional Comments:**

The current content of the paper is fine. Format, however, should be fixed, both by removing colours and strikethrough text.

**Audience:**

Yes

**Audience Explanation:**

Graph generation is an active and rapidly evolving research area, with strong potential for impact across several important domains, including drug design.

**Claims And Evidence:**

Yes

**Claims Explanation:**

The revised version of the paper, after constructively  considering the comments by reviewers, provides strong and clear evidence that incorporating graph filtrations into autoregressive generation enables the proposed approach to address generation errors through the modeling of non-monotonic graph sequences. The primary advantage of the proposed method lies in its ability to achieve a more favorable trade-off between quality and inference time compared to competing approaches.
All principal concerns raised by the reviewers have been thoroughly addressed, and the current version of the paper is satisfactory.

---

> ### Author Response · Authors · 2026-01-22
>
> Dear Action Editor,
>
> We fixed the format and uploaded the camera-ready version. Please note that we have removed the "Towards" in the title as we find it more correct to describe our method.
>
> Thank you again for your time and handling of our submission.
>
> Best regards,
>
> The authors